# PTE4TS: One Pre-Training Encoder is All Time Series Need

## Abstract

In Natural Language Processing (NLP) and Computer Vision (CV) as well as myriad other domains, pre-trained models have achieved significant breakthroughs. However, their advancements in the sphere of general Time Series Analysis (TSA) has been comparatively limited. The principal challenge lies in the dearth of extensive training data that is endemic to the field of TSA. This scarcity hampers the direct application of such pre-training models to time series data, resulting in unsatisfactory performance. Despite numerous attempts to adapt NLP or CV models, which have been pre-training on billions of tokens, to TSA to address this challenge, these pre-training models are not directly suitable for time series data. In this work, we introduce a new general Pre-Training Encoder specifically designed for Time Series analysis, called PTE4TS. It is designed to be universal for any type of time series data, and is easy to adapt to various downstream tasks such as classification, anomaly detection, and forecasting. First, we revisited the masking methods in time series and found that patch masking, which was widely adopted previously, is not necessary. Therefore, we developed an improved masking model tailored to the characteristics of time series. Additionally, to address the issue of the Low-Rank structure in conventional bidirectional attention mechanisms, which may diminish the model's expressiveness, we have developed a straightforward yet efficacious hybrid attention encoder. The combination of this encoder with our masking methods can improve the representation ability of the model. Finally, PTE4TS achieved state-of-the-art performance on several real-world datasets, further validating the potential of Large Model for general time series analysis. We hope that PTE4TS will not only open new perspectives in the field of TSA, enhancing feature representation and inferencing capabilities across various domains, but also lay the foundation for a general artificial intelligence that is capable of understanding and processing common time series data.

## 1 Introduction

The disciplines of Natural Language Processing (NLP) Radford et al. (2018); Devlin et al. (2018) and Computer Vision (CV) He et al. (2022); Feichtenhofer et al. (2022) have witnessed a surge in innovation thanks to the development of Large pre-training Models, propelling both fields into an era of breakthroughs Zhao et al. (2023); Awais et al. (2023). However, the impact of these large models within the specialized realm of Time Series Analysis (TSA) has been notably subdued. A significant factor contributing to this disparity is the deficit of large-scale, versatile training datasets tailored to TSA, which compromises the efficacy of these sophisticated models when applied to the distinctive nature of time series data Zhou et al. (2023). The scientific community has made concerted efforts to transpose the success of pre-training NLP and CV models, models that have been honed on extensive corpora of billions of tokens, to the context of TSA Liu et al. (2023a); Yu et al. (2023); Gruver et al. (2023). Nevertheless, the challenge remains formidable. The underlying architecture and training of these models are not inherently compatible with the unique characteristics of time series data. Therefore, while NLP and CV benefit from transformative advances owing to these large models, the quest continues to either cultivate TSA-specific models or ingeniously modify existing pre-training ones to resonate with the complex patterns and dynamics inherent in time series analysisLiu et al. (2024).

Time series data is data collected or recorded at multiple points in time, and it is often utilized to observe or predict phenomena that change over time. This type of data is widely applied across various fields such as electricity Zhou et al. (2021), weather Wu et al. (2021) , transportation Yin & Shang (2016), disease prevalence Liu et al. (2018), and more. Time series data can vary significantly in dimensions, length, and sampling frequency depending on the field of application, presenting considerable challenges for data integration and cross-disciplinary research. It can be difficult to align these diverse datasets for training within a single model. Furthermore, since Time series data typically requires long-term accumulation, some specialized areas may experience a shortage of data, insufficient for the needs of training complex models Godahewa et al. (2021). The mask task technique can help mitigate issues related to data scarcity. The fundamental concept behind this technique is to randomly mask portions of the data within the training set, compelling the model to learn in the absence of some information, thereby enhancing the model's adaptability to unseen data Devlin et al. (2018); He et al. (2022). This approach simulates a more diverse data environment by presenting slightly different inputs in each iteration of training. Randomly masking the training data creates a scenario where the input is likely to be different each time, and a high masking ratio can heighten this probability.

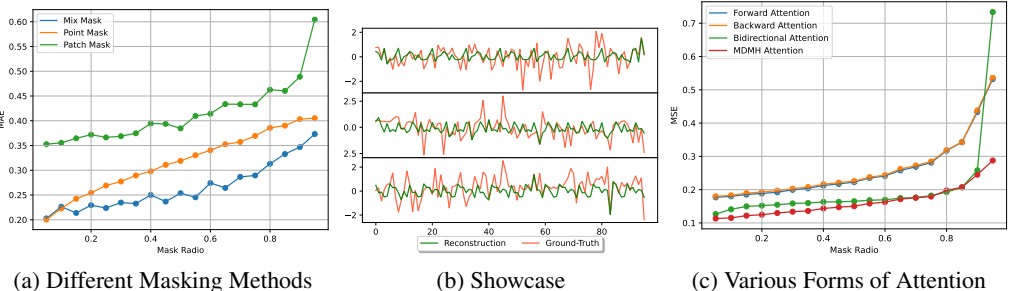

| (a) Different Masking Methods | (b) Showcase | (c) Various Forms of Attention |

Figure 1: (a)The results of the Mean Absolute Error (MAE) within the ETTh2 dataset after applying bidirectional Attention to inputs that have been obscured at various ratios are discussed. Herein, "Mix" represents the method proposed by us. It is evident that in reconstruction tasks, our method demonstrates superior detailed performance compared to both Patch-Level and Point-Level approaches. (b) The output from the Bidirectional Attention after the input has been masked at Patch-Level. The Reconstruction results showed obvious repetitive patterns that do not exist in Ground Truth. (c) The Mean Squared Error (MSE) outcomes on the Electricity (ECL) dataset subsequent to the application of various proportions of masking on the input at the Patch-Level, followed by the implementation of distinct types of Attention. Notably, MDMH represents the methodology we have introduced.

However, in the domain of time series analysis, the masking approach within mask models merely adopts methodologies from CV, aggregating time steps into sub-sequence level patches and implementing masking at the patch level Tang & Zhang (2022); Nie et al. (2022). However, these methods fail to consider the unique characteristics of time series data and disregard point-level information. Furthermore, the commonly utilized bidirectional Attention mechanism in mask models suffers from the issue of a low-rank structure, which, to a certain extent, restricts the model's representational capacity. This reduction in representational capacity diminishes the model's ability to capture complex dependencies between input data, ultimately leading to a loss of information Bhojanapalli et al. (2020); Dong et al. (2021). However, in the widely used decoder-only architectures of large models, the unidirectional attention lacks bidirectional interaction capabilities, and the autoregressive generation which predicts the next token step-by-step requires more time and computational resources.

Motivated by the aspects discussed above, we have introduced the PTE4TS model as a novel, general representation for TSA. The pioneering aspect of this model lies in its unique approach to pre-training, which deviates from the traditional methods that bypass pre-training and go straight to fine-tuning using Large Language Model (LLMs). Specifically, our contributions can be summarized into three aspects:

- We developed a specialized pre-training strategy specifically tailored for time series data, departing from the conventional, simplistic patch-level masking approaches. Instead, we

have combined point-level and patch-level processing techniques. This integrative approach enhances our model's ability to effectively learn and capture local features and patterns within time series data. Furthermore, our strategy is adept at accommodating time series of varying lengths and easily adapts to datasets of different scales.

- To address the diminishing expressive capabilities arising from the Low-Rank structures of bidirectional attention, and considering the limitations of unidirectional attention in handling masked models, we created a hybrid attention encoder that is both straightforward and efficient. This innovative encoder adeptly tackles the aforementioned issues, proving to be highly effective in the context of masked models.

- Our PTE4TS model has achieved outstanding or competitive state-of-the-art performance across all mainstream time series analysis tasks. These tasks span time series classification, long-term forecasting, data imputation, and anomaly detection, among others. The widespread and comprehensive experimental underpinning the model's performance further substantiates the effectiveness of our proposed methods and its enormous potential for widespread application.

## 2 PRELIMINARIES

### 2.1 MASKING METHODS

In our exploration of the TSA masking models, we've focused on the PatchTST method . In the mask model of time series analysis, most methods opt to borrow from the BERT Devlin et al. (2018) in NLP and the MAE He et al. (2022) in CV. These approaches are predominantly based on the patch-based Transformer model. They do not significantly alter the Transformer; instead, they simply add a patch method or, in some cases, replace the original decoder with a single layer of MLP Nie et al. (2022); Tang & Zhang (2022). Patch can compress data and reduce the dimensionality of input data, to a certain extent lowering redundant features. Moreover, patch have a smoothing effect that can mitigate the impact of outliers and help filter fluctuations and random noise, retaining only more stable and representative information.

However, as illustrated in Figure 1a, we've identified certain limitations in their application. While patch-level masking does create a more challenging task for the model by increasing the difficulty of reconstructing the masked data, which can help the model learn local features of the data, this technique can also lead the model to overemphasize recurrent patterns ( Figure 1b) and neglecting continuous changes between patches, resulting in difficulty in learning the "bigger picture", such as trends. In Figure 1b, we have selected three visualization cases to more clearly demonstrate how patch-level masking leads the model to overly focus on repeated patterns. The reconstruction results in the figure clearly show that the model's output consists almost entirely of repeated patterns, indicating that the model has not learned detailed local information. On the other hand, tasks with a higher point-level mask ratio are not only capable of constructing a similar level of difficulty but also ensure that the model remains sensitive to minor variations. This enables the model to better learn the original semantics without overlooking the point-level features. Furthermore, due to its increased uncertainty, point-level masking provides a unique learning environment for the model, emphasizing a deeper understanding of the data's intrinsic structure, going beyond simple pattern recognition.

### 2.2 LOW-RANK IN ATTENTION

In the field of NLP, the construction and optimization of LLMs stand as a key topic of discussion. These models are typically designed with a Decode-only architecture Radford et al. (2019); Brown et al. (2020), which has shown significant performance enhancements over Encode-only methods in multiple studies. However, the Decode-only structure is not always the optimal choice for every task. We have a particular interest in the role of the Attention mechanism within mask task. As depicted in Figure 1c, the unidirectional Attention mechanism inherent in Decode-only architectures falls short on mask task, while bidirectional Attention, which can potentially perform better, faces its own set of issues. Specifically, the problems with bidirectional Attention often boil down to the Low-Rank structure it exhibit. In the Attention mechanism, the Attention matrix is usually produced through a process of Low-Rank matrix factorization-typically, by dot-product such as $L \times D$ with

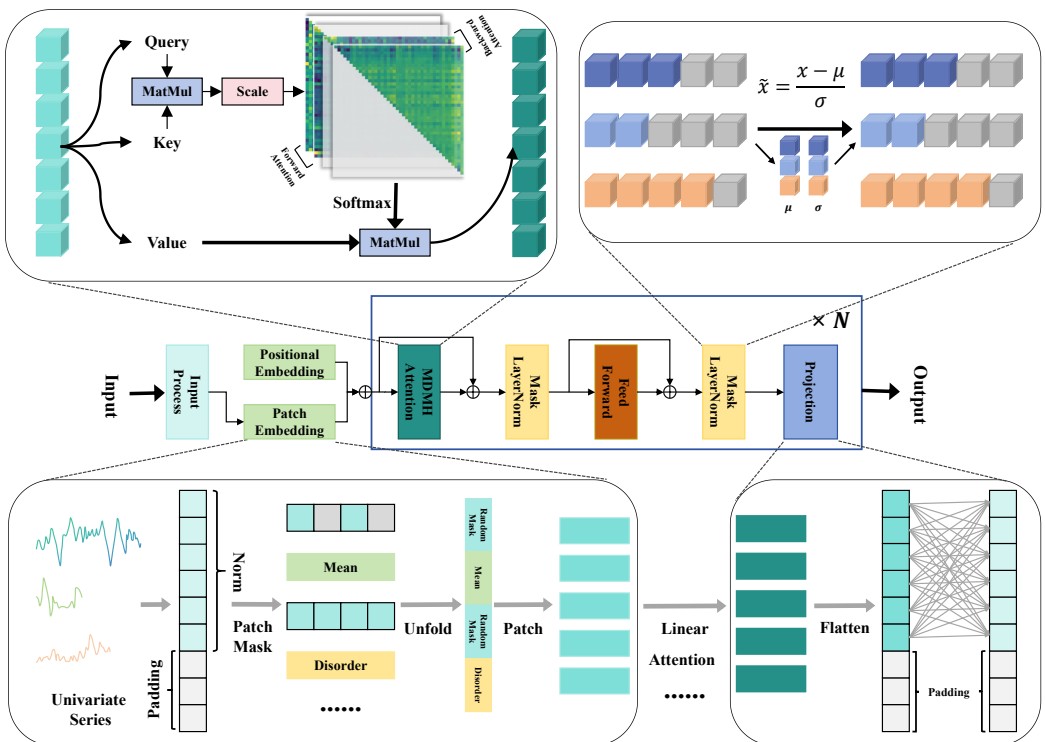

Figure 2: Overall structure of PTE4TS. First, multivariate time series from different sources are treated as univariate time series for separate processing, and then padded to the same length. Next, the processed sequences are patched into tokens, and Embedding is implemented using a multi-layer perceptron (MLP). Then, self-attention in various directions is applied to the embedded variable tokens, allowing for better context consideration and the capture of complex dependencies. Finally, a projection maps the latent vectors back to patch-level tokens, followed by an MLP that enhances point-level interactions.

$D \times L$ (usually $L$ is the input length, $D$ is the hidden size, $D < L$), and then normalizing the resultant matrix with a softmax function. Due to its Low-Rank nature, the resulting Attention matrix may, to a certain degree, diminish the model's representational capacity. On the other hand, under the Decode-only architecture, the Attention matrix presents as a lower-triangular matrix. Owing to the characteristics of the softmax function, which ensures that the elements along the diagonal are positive, the determinant of this matrix is non-zero, indicating that such an Attention matrix is Full-Rank. Theoretically, a Full-Rank Attention matrix possesses a broader expressive capability to capture subtle dependencies across a wider range of tasks and contexts. Because of this Full-Rank feature, from a theoretical standpoint, the Decode-only architecture offers a more robust learning framework, particularly adept at handling sequence generation tasks where leveraging historical information is crucial. Nevertheless, this doesn't mean that bidirectional attention lacks practical value. In fact, for many feature extraction or sequence understanding tasks, bidirectional attention is indispensable due to its ability to integrate context information. Therefore, a potential research direction is to explore the design of a new Attention mechanism that could either sparsify attention or combine the representational strength of the Decode-only architecture with the contextual integration capability of bidirectional attention.

## 3 PTE4TS

In multivariate time series analysis, it is common to be given historical observations $\mathbf{X}^N = \{x_1, \cdots, x_L\} \in \mathbb{R}^L$ from $L$ time steps, composed of $N$ univariate time series $\mathbf{X}$. Clearly, multivariate time series tasks can also be decomposed into univariate time series tasks. Under this condition, we are only concerned with the feature within each univariate series, while ignoring the interactions

across different variables. One of the main reasons for this is that in real-world scenarios, it is possible that the time points do not contain exactly the same timestamps Liu et al. (2023b), and enforcing cross-variable alignment in the model can easily lead to the learning of chaotic systems.

We propose the PTE4TS as shown in Figure 2, which utilizes a Transformer Vaswani et al. (2017) architecture that can be viewed as an Encoder-only structure, but with some simple modifications. It additionally includes an input process, embedding, and projection. Similar to the previous Mask Model, we do not use redundant Encoder-Decoder for representation learning. The task of generating complete series can essentially be handed over to a linear layer, which has also been demonstrated in previous work, a simple linear layer is sufficient for many tasks Zeng et al. (2023); Chen et al. (2023).

Our PTE4TS focuses on representation learning and adaptive association. Each time series, driven by potentially complex processes, is first patched to describe the nature of local features. They are then subjected to mutual interactions via attention, and processed into patch representations by a feedforward network. After multiple layers of the Encoder-Block, the original input is gradually reconstructed.

## 3.1 INPUT PROCESS AND EMBEDDING

To better focus on point-level attention and create a more challenging masking task, we need to redesign previous masking methods to suit the characteristics of time series data. First, we apply instance normalization to the input data to facilitate knowledge transfer. Next, we partition the time series data into patches. In this stage, we select patches to be masked at the patch level but instead of simply initializing these parts randomly, we assign the mean value of the patch or randomly shuffle the order within the patch. We hope this type of masking will enable the model to learn point-level sequential information within the series, as well as the relationship between the original data and its mean. For the remaining data that still needs to be masked, we perform point-level masking, with the selected points being randomly initialized.

Time series data is typically recorded at successive time points and then input into the model. However, in PatchTST, adjacent and continuous time points are treated as a patch before being input into the model. For point masking, each time point is considered an individual entity to be masked, preserving the trend of the time series. In contrast, patch masking treats adjacent and continuous time points as an entire patch to be masked. This means that primarily consecutive time points are masked, resulting in the remaining data lacking a continuous trend.

Our specific averaging operation targets the part that needs to be masked by setting this portion of the data to the mean of the remaining visible data. Random shuffling is also targeted at the part that needs to be masked, shuffling this portion within the patch. This approach helps the model learn the relationships between data points.

The input process is responsible for processing time series data into the form required for pre-training. Subsequently, an embedding projects the data onto the dimensions needed by the pre-training model. We have chosen the patch embedding method, noting that the patch size in patch embedding and the patch size in the patch mask can differ. Patch embedding, by aggregating information from adjacent time points, forms a patch-based token that can better extract local semantic information, significantly increasing the temporal scope of the input data and reducing informational redundancy in the Transformer model.

## 3.2 MIX ATTENTION ENCODER BLOCK

The original scaled dot-product attention mechanism performs as:

$$Attention(\mathbf{Q}, \mathbf{K}, \mathbf{V}) = Softmax(\frac{\mathbf{Q}\mathbf{K}^T}{\sqrt{d}})\mathbf{V} \tag{1}$$

i.e., *Attention* is defined as an operation of ternary matrix, where $\mathbf{Q}(queries) \in \mathbb{R}^{L_Q \times d}$, $\mathbf{K}(keys) \in \mathbb{R}^{L_K \times d}$, $\mathbf{V}(values) \in \mathbb{R}^{L_V \times d}$, and $d$ is the feature dimension. Typically, $L$ is greater than $d$, resulting in Attention matrices that often have a Low-Rank structure, which can reduce the representational capacity of the model to some extent.

Furthermore, we consider unidirectional Attention. Unidirectional Attention matrices take the form of a triangular matrix, and due to the properties of the softmax function, it ensures that the elements along the diagonal are all positive, which means the unidirectional Attention matrix is Full-Rank. Theoretically, a Full-Rank Attention matrix possesses a more diversified expressive capacity, hence it can capture subtle dependencies in a wider range of tasks and scenarios, providing a more powerful learning framework. However, for mask tasks, the masked section should consider not only historical information (Forward Attention) but also future information (Backward Attention) effectively. Therefore, we extend unidirectional Attention to a multi-head level, truncating half of the Attention heads into lower triangular matrices (Forward Attention, FA), and the other half into upper triangular matrices (Backward Attention, BA).

Based on the above considerations, Mixed-Direction Multi-Head (MDMH) Attention can be simply expressed as follows:

$$FH_i(Q, K, V) = FA(QW_i^Q, KW_i^K, VW_i^V)$$
$$BH_i(Q, K, V) = BA(QW_i^Q, KW_i^K, VW_i^V) \tag{2}$$
$$MDMH(Q, K, V) = Concat(FH, BH)W^O$$

Where the projections are parameter matrices $W_i^Q, W_i^K, W_i^V \in \mathbb{R}^{d_{model} \times d}$, and $W^O \in \mathbb{R}^{hd \times d_{model}}$, $0 \le i < h/2$, $h$ is the number of heads. This segmented-style unidirectional attention helps the model to better capture causal relationships and temporal patterns in time series data. Finally, we combine Multi-Direction Multi-Head (MDMH) Attention with Feedforward Neural Network (FFN) to ensure that the model effectively considers the Forward-Backward interaction, taking into account the context comprehensively and capturing complex dependencies.

### 3.3 MASKED LAYERNORM

Layer normalization (LN) Ba et al. (2016) is a normalization technique commonly used in deep neural networks to mitigate the issue of internal covariate shift during the training process. In typical LN, the module computes the mean and variance of all the features within each sample of the input to that layer. However, for the training data of PTE4TS, in order to make the sequences uniform in length, we apply padding to supplement the data from shorter sequences, which may result in the effective data length being much shorter than the length of the input data. These padding values do not represent effective information in reality, and including these padding parts when calculating the mean and variance could mistakenly incorporate irrelevant noise into the model's learning process, leading to decreased performance and slower convergence. To address this issue, we have designed a variant called Masked LayerNorm. In this version, we use a mask to identify the padding parts within the input data. Masked LayerNorm first calculates the mean $\mu$ and variance $\sigma^2$ across all the outputs of the layer:

$$\mu = \frac{1}{\sum m} \sum_{i=1}^{H} (m_i \cdot x_i)$$
$$\sigma^2 = \frac{1}{\sum m} \sum_{i=1}^{H} m_i \cdot (x_i - \mu)^2 \tag{3}$$

The output x from a certain layer in the network has dimensions $[d_1, ..., d_k]$, where $d_i$ represents the size of dimension i. The corresponding mask for x is denoted as m, with $m_i$ being 0 indicating the padding part. When calculating the mean and variance, we only consider the valid, non-padded data to effectively avoid the interference of noise on the normalization process. Then, we use the computed mean and variance to normalize the layer's output. After normalization, we introduce two learnable parameters, $\gamma$ (scale) and $\beta$ (shift), so that the model has the ability to restore the original representation of the data.

$$\hat{x}_i = \frac{x_i - \mu}{\sqrt{\sigma^2 + \epsilon}}$$
$$y_i = \gamma \hat{x}_i + \beta \tag{4}$$

Here, $\epsilon$ is a very small number added for numerical stability to prevent division by zero. Usually, the value of $\epsilon$ is set between $10^{-5}$ and $10^{-8}$. The final output is $y_i$, while $\gamma$ and $\beta$ are the trained

parameters whose shapes are consistent with the dimensions of $x$. With this approach, our model can not only more accurately reflect the statistical properties of the true features when processing time series data, but it can also significantly improve the robustness and performance of the model when dealing with sequences of different lengths.

# 4 EXPERIMENTS

We pre-trained our model on the Unified Time Series Dataset (UTSD) Liu et al. (2024) and then applied the model to unseen new time series data (ETTH, ECL, Traffic, Weather, PEMS03, PEMS04 and so on). UTSD is a dataset containing up to 1 billion time points across seven domains, with all datasets categorized into seven distinct domains based on their sources: Energy, Environment, Health, Internet of Things (IoT), Nature, Traffic, and Web, and featuring diverse sampling frequencies. We selected UTSD-12G as our pre-training dataset.

Our proposed method exhibits excellent performance across a range of TSA tasks through fine-tuning. To demonstrate the effectiveness of our approach, we conducted extensive experiments on primary types of downstream tasks, including time series classification, anomaly detection, imputaion, and long-term forecasting. To ensure a fair comparison, we adhered to the experimental setup of TimesNet Wu et al. (2022), and the main experimental results were derived from FPT Zhou et al. (2023). Although there is some overlap between our pre-training tasks and downstream tasks in terms of datasets, to prevent data leakage, we only use the training set portion of these datasets for any training, ensuring that the test set and validation set remain unseen during the training phase. We provide detailed implementations and model configurations for pre-training and fine-tuning in the appendix.

**Baselines:** We have selected a representative baseline and cited the results from FPT Zhou et al. (2023). The baselines include CNN-based models: TimesNet Wu et al. (2022); MLP-based models: LightTS Zhang et al. (2022) and DLinear Zeng et al. (2023); Transformer-based models: Time Liu et al. (2024), FPT Zhou et al. (2023), Autoformer Wu et al. (2021), FEDformer Zhou et al. (2022), Non-stationary Transformer Liu et al. (2022), ETSformer Woo et al. (2022), PatchTST Nie et al. (2022). Besides, Anomaly Transformer Xu et al. (2021) is used for anomaly detection. XGBoost Chen & Guestrin (2016), Rocket Dempster et al. (2020), LSTNet Lai et al. (2018), LSSL Gu et al. (2021), Pyraformer Liu et al. (2021), TCN Franceschi et al. (2019) and Flowformer Huang et al. (2022) are used for classification.

## 4.1 FORECASTING

Time series forecasting involves analyzing data points that evolve over time to predict future trends at a specific timestep or over a period. This process entails forecasting the upcoming values Y of a univariate time series for T timesteps ahead, based on the known values X. For long-term forecasts, we expect T to exceed a certain threshold L. In the fine-tuning phase, we replaced the Projection module with a simple linear layer, which has been proven to perform well in time series forecasting. Subsequently, we maintained all other components unchanged and proceeded with fine-tuning on various datasets separately.

### 4.1.1 FEW-SHOT FORECASTING

To better compare with Timer, we set the context length for pre-training to 1440 and the lookback length for downstream prediction tasks to 672. The final experimental results are shown as Figure 3, demonstrating the powerful performance and impressive generalization capabilities of PTE4TS on previously unseen time series data. Based on the pre-trained PTE4TS model, we gradually reduce the training data used for fine-tuning in downstream tasks, from the full amount of 100% down to just 1%. We observe that the amount of fine-tuning data does not significantly impact PTE4TS. Overall, PTE4TS exhibits a trend of slow performance decline as training data decreases, without any sudden drops in performance. This indicates that for all time series downstream tasks, a pre-trained model can achieve ideal performance with minimal training.

We observed that the amount of fine-tuning data does not significantly impact PTE4TS. Overall, PTE4TS exhibits a trend of slow performance decline as training data decreases, without any sudden

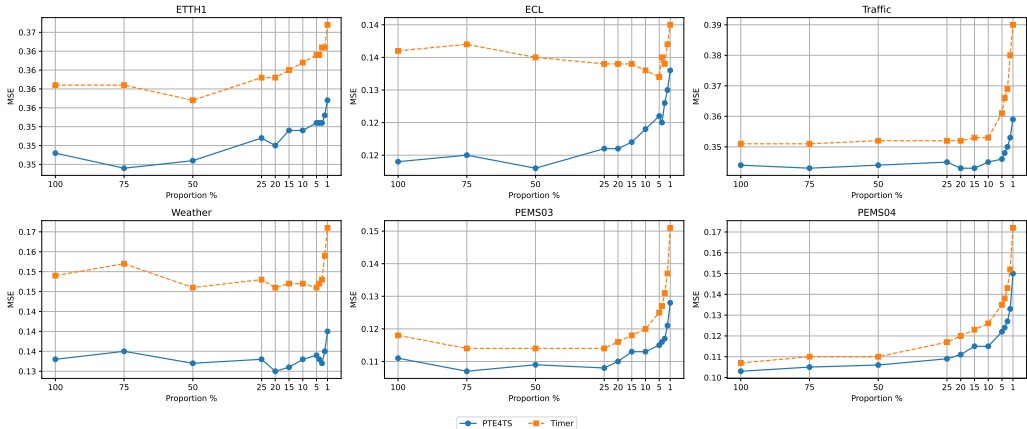

Figure 3: Results of PTE4TS and Timer on downstream forecasting tasks obtained by fine-tuning from the pre-trained model.

drops in performance. In twelve setups across six datasets, we achieved impressive SOTA results. It can be seen that PTE4TS was 15.66% lower on the Weather dataset at 1% proportion and 15.23% lower on the PEMS03 dataset at 1% proportion. This strongly demonstrates the powerful few-shot capability of PTE4TS. This indicates that for all time series downstream tasks, a pre-trained model can achieve ideal performance with minimal training.

### 4.1.2 LONG-TERM FORECASTING

Table 1: Long-term forecasting task. All the results are averaged from 4 different prediction lengths, that is {24, 36, 48, 60} for ILI and {96, 192, 336, 720} for the others. We follow the configurations of Dlinear, in which the sequence length for the past in ILI is set to 104, while for the other sequences it is 336. Other models adhere to the results presented in the FPT paper. The best results are represented in **bold**, and the appendix displays the complete results.

| Methods | PTE4TS | | FPT | | TimesNet | | PatchTST | | DLinear | | ETSformer | | LightTS | | FEDformer | | Non-Stationary | | Autoformer | |
|---|---|---|---|---|---|---|---|---|---|---|---|---|---|---|---|---|---|---|---|---|
| Metric | MSE | MAE | MSE | MAE | MSE | MAE | MSE | MAE | MSE | MAE | MSE | MAE | MSE | MAE | MSE | MAE | MSE | MAE | MSE | MAE |
| ETTh$_1$ | **0.404** | **0.418** | 0.427 | 0.426 | 0.458 | 0.450 | 0.413 | 0.430 | 0.423 | 0.437 | 0.542 | 0.510 | 0.491 | 0.479 | 0.440 | 0.460 | 0.570 | 0.537 | 0.496 | 0.487 |
| ETTh$_2$ | **0.322** | **0.368** | 0.346 | 0.394 | 0.414 | 0.427 | 0.330 | 0.379 | 0.431 | 0.447 | 0.439 | 0.452 | 0.602 | 0.543 | 0.437 | 0.449 | 0.526 | 0.516 | 0.450 | 0.459 |
| ETTm$_1$ | **0.328** | **0.360** | 0.352 | 0.383 | 0.400 | 0.406 | 0.351 | 0.387 | 0.357 | 0.378 | 0.429 | 0.425 | 0.435 | 0.437 | 0.448 | 0.452 | 0.481 | 0.456 | 0.588 | 0.517 |
| ETTm$_2$ | **0.246** | **0.303** | 0.266 | 0.326 | 0.291 | 0.333 | 0.255 | 0.315 | 0.267 | 0.334 | 0.293 | 0.342 | 0.409 | 0.436 | 0.305 | 0.349 | 0.306 | 0.347 | 0.327 | 0.371 |
| Weather | **0.215** | **0.251** | 0.237 | 0.270 | 0.259 | 0.287 | 0.225 | 0.264 | 0.249 | 0.300 | 0.271 | 0.334 | 0.261 | 0.312 | 0.309 | 0.360 | 0.288 | 0.314 | 0.338 | 0.382 |
| ECL | **0.145** | **0.238** | 0.167 | 0.263 | 0.192 | 0.295 | 0.161 | 0.253 | 0.166 | 0.263 | 0.208 | 0.323 | 0.229 | 0.329 | 0.214 | 0.327 | 0.193 | 0.296 | 0.227 | 0.338 |
| Traffic | **0.382** | **0.255** | 0.414 | 0.294 | 0.620 | 0.336 | 0.390 | 0.264 | 0.434 | 0.295 | 0.621 | 0.396 | 0.622 | 0.392 | 0.610 | 0.376 | 0.624 | 0.340 | 0.628 | 0.379 |
| ILI | **1.324** | **0.721** | 1.925 | 0.903 | 2.139 | 0.931 | 1.443 | 0.798 | 2.169 | 1.041 | 2.497 | 1.004 | 7.382 | 2.003 | 2.847 | 1.144 | 2.077 | 0.914 | 3.006 | 1.101 |

To assess long-term forecasting, we tested our models on eight popular real-world datasets including Electricity, Weather, Traffic, ILI, and the four ETT datasets. In the realm of long-term time series forecasting, the approach utilizing patches has been validated to exhibit commendable performance Nie et al. (2022); Gong et al. (2023); Ekambaram et al. (2023). This technique, including the application in the FPT, adopts a strategy of patching. Our model perpetuates this strategy by applying patches to the input data. This methodology effectively enables the model to enhance its learning of local information and concurrently abbreviates the length of the input sequence. Such an approach significantly ameliorates the model's capability to manage long-range dependencies more efficiently. Table 1 presents the average results for multiple prediction baselines across each dataset, where it is evident that PTE4TS consistently achieved the best outcomes. Conversely, for the FPT, which bypassed pre-training and was directly fine-tuned on GPT, the forecasting performance did not surpass that of PatchTST. This observation clearly suggests that the straightforward approach of fine-tuning on large language models is not particularly effective.

## 4.2 IMPUTATION

In practical applications, time series data can often exhibit gaps due to a variety of reasons, such as sensor malfunction, data loss, or transmission errors. Time series data is an array of data points arranged in chronological order, and the effectiveness of their analysis and forecasting hinges on their integrity and continuity. Therefore, filling in missing values is of great significance. In the task of time series imputation for each univariate time series, our usual input, X, is incomplete and contains missing values. The goal of imputation is to restore the data to its original, complete state. Our experiments spanned across six commonly used,real-world datasets, including four ETT datasets, Electricity and Weather datasets, where missing data occurrences are not uncommon. To benchmark the performance of our models across different missing data scenarios, we followed the TimesNet setup and chose varying masking ratios of 12.5%, 25%, 37.5%, and 50% to randomly mask timestamps, simulating different levels of data incompleteness.

Table 2: Imputation task. We perform a randomly mask on the time series with an input length of 96 at {12.5%, 25%, 37.5%, 50%}. The results are averaged from 4 different mask ratios. The best performance is indicated in **Blod**, and the appendix presents the complete findings.

| Methods | PTE4TS | | FPT | | TimesNet | | PatchTST | | ETSformer | | LightTS | | DLinear | | FEDformer | | Non-Stationary | | Autoformer | |
|---|---|---|---|---|---|---|---|---|---|---|---|---|---|---|---|---|---|---|---|---|
| Metric | MSE | MAE | MSE | MAE | MSE | MAE | MSE | MAE | MSE | MAE | MSE | MAE | MSE | MAE | MSE | MAE | MSE | MAE | MSE | MAE |
| ETTm$_1$ | **0.021** | **0.096** | 0.028 | 0.105 | 0.027 | 0.107 | 0.047 | 0.140 | 0.120 | 0.253 | 0.104 | 0.218 | 0.093 | 0.206 | 0.062 | 0.177 | 0.036 | 0.126 | 0.051 | 0.150 |
| ETTm$_2$ | **0.015** | **0.075** | 0.021 | 0.084 | 0.022 | 0.088 | 0.029 | 0.102 | 0.208 | 0.327 | 0.046 | 0.151 | 0.096 | 0.208 | 0.101 | 0.215 | 0.026 | 0.099 | 0.029 | 0.105 |
| ETTh$_1$ | **0.061** | **0.164** | 0.069 | 0.173 | 0.078 | 0.187 | 0.115 | 0.224 | 0.202 | 0.329 | 0.284 | 0.373 | 0.201 | 0.305 | 0.17 | 0.246 | 0.094 | 0.201 | 0.103 | 0.314 |
| ETTh$_2$ | **0.040** | **0.131** | 0.069 | 0.173 | 0.078 | 0.187 | 0.115 | 0.224 | 0.202 | 0.329 | 0.284 | 0.373 | 0.201 | 0.306 | 0.117 | 0.246 | 0.094 | 0.201 | 0.103 | 0.214 |
| ECL | **0.061** | **0.173** | 0.090 | 0.207 | 0.092 | 0.210 | 0.072 | 0.183 | 0.214 | 0.339 | 0.131 | 0.262 | 0.132 | 0.260 | 0.130 | 0.259 | 0.100 | 0.218 | 0.101 | 0.225 |
| Weather | **0.023** | **0.046** | 0.031 | 0.056 | 0.030 | 0.054 | 0.034 | 0.055 | 0.076 | 0.171 | 0.055 | 0.117 | 0.052 | 0.110 | 0.099 | 0.203 | 0.032 | 0.059 | 0.031 | 0.057 |

The imputation task bears resemblance to the reconstruction task during the pre-training phase, yet we have selected a higher mask ratio (75%) during the pre-training phase. The extremely high masking ratio constructs a task that is challenging to recover from, making it difficult for the model to restore the masked information from adjacent temporal points. Such a task enhances the model's comprehension beyond mere low-level information, fostering an understanding of deeper semantics. Owing to the similarity of the tasks, we opted not to replace the Projection module during the fine-tuning phase. Table 2 shows that PTE4TS achieved the best results on all data.

## 4.3 ANOMALY DETECTION

The task of time series anomaly detection involves identifying outliers or anomalous segments within time series data, which deviate from expected patterns and could be indicative of underlying issues or significant events. Our study involved a comparative analysis of models on five standard datasets: SMD, MSL, SMAP, SWaT, and PSM. To ensure a level playing field, all baseline models were confined to utilizing reconstruction errors, with data preprocessing methods and model configurations keeping consistent with those employed in the TimesNet approach.

Table 3 presents the F1-scores on five datasets, revealing that PTE4TS still achieves consistent state-of-the-art performance in this challenging task. This indicates that employing the patching method on the input not only enhances the model's ability to learn local features but also does not interfere with the detection of anomalies.

## 4.4 CLASSIFICATION

In data science, time series classification refers to the process of pattern recognition by analyzing sequences of data points that are organized chronologically, and subsequently assigning these sequences to predefined categories. This technique has found broad applications across various fields, such as financial forecasting where it's used to predict stock price movements, in healthcare for analyzing electrocardiograms, in industry for detecting machinery faults, and in meteorology for weather forecasting. To assess the performance in this context, we selected 10 diverse UEA multivariate classification datasets and preprocessed them following the approach used in TimesNet, with each subset having varying lengths of sequences.

Table 3: Anomaly detection task. We calculate the F1-score (as %) for each dataset. **Blod** represents the best, and the appendix displays the complete results.

| Methods | SMD | MSL | SMAP | SWaT | PSM | Average |
|---|---|---|---|---|---|---|
| PTE4TS | **87.76** | **89.44** | **75.31** | **94.75** | **97.70** | **88.99** |
| FPT | 86.89 | 82.45 | 72.88 | 94.23 | 97.13 | 86.72 |
| TimesNet | 84.61 | 81.84 | 69.39 | 93.02 | 97.34 | 85.24 |
| PatchTST | 84.62 | 78.70 | 68.82 | 85.72 | 96.08 | 82.79 |
| ETSformer | 83.13 | 85.03 | 69.50 | 84.91 | 91.76 | 82.87 |
| FEDformer | 85.08 | 78.57 | 70.76 | 93.19 | 97.23 | 84.97 |
| LightTS | 82.53 | 78.85 | 69.21 | 93.33 | 97.15 | 84.23 |
| DLinear | 77.10 | 84.88 | 69.26 | 87.52 | 93.55 | 82.46 |
| Non-Stationary | 84.72 | 77.50 | 71.09 | 79.88 | 97.29 | 82.08 |
| Autoformer | 85.11 | 79.05 | 71.12 | 92.74 | 93.29 | 84.26 |
| Pyraformer | 83.04 | 84.86 | 71.09 | 91.78 | 82.08 | 82.57 |
| Anomaly Transformer | 85.49 | 83.31 | 71.18 | 83.10 | 79.40 | 80.50 |
| Informer | 81.65 | 84.06 | 69.92 | 81.43 | 77.10 | 78.83 |
| Reformer | 75.32 | 84.40 | 70.40 | 82.80 | 73.61 | 77.31 |
| LogTrans. | 76.21 | 79.57 | 69.97 | 80.52 | 76.74 | 76.60 |
| Transformer | 79.56 | 78.68 | 69.70 | 80.37 | 76.07 | 76.88 |

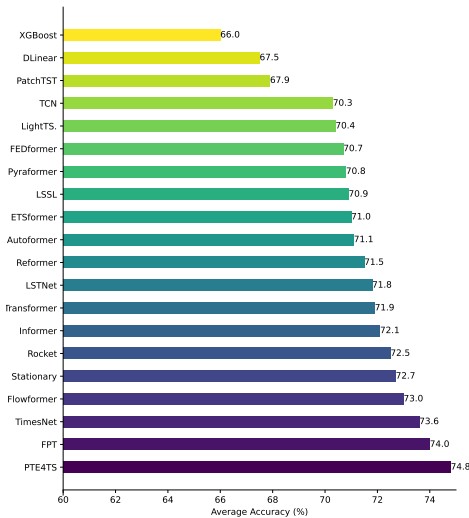

Figure 4: Model comparison in classification.

The span of sequence lengths within the categorized datasets is considerable, ranging from 29 to 1751. To facilitate more effective application of PTE4TS, we have not only substituted the Projection module with a simple linear layer for categorization purposes, but we have also adapted the patch-size for each dataset to ensure compatibility with the input requirements of PTE4TS. Unlike forecasting and anomaly detection tasks, the datasets employed for classification have not been exposed to the model during the pre-training phase; however, this does not impede the performance of PTE4TS. As illustrated in Figure 4, it is evident that PTE4TS achieves the highest mean accuracy rates across the 10 datasets under examination.

## 5 CONCLUSION

The PTE4TS model proposed in this paper marks an innovative breakthrough in the field of time series analysis. By adopting a novel pre-training strategy, PTE4TS surpasses traditional methods, becoming the first general time series analysis model pre-trained from scratch, effectively solving key issues in processing time series data. Our model not only brings theoretical innovation but also confirms its superior performance through a series of experiments in practical applications, demonstrating the effectiveness and vast application potential of our approach. The core innovation of the PTE4TS model lies in combining point-level and patch-level mask pre-training methods, which are more conducive to learning local features and overall patterns in time series data. Moreover, to overcome the limitations of Low-Rank in bidirectional attention and the shortcomings of unidirectional attention in mask models, we designed an efficient hybrid attention encoder that optimizes the model's expressive capabilities when dealing with masking issues. In various mainstream time series analysis tasks, including time series classification, long-term forecasting, data imputation, and anomaly detection, the PTE4TS model exhibits outstanding performance, on par with or surpassing existing state-of-the-art technology. Extensive experimental results demonstrate the versatility and efficiency of the PTE4TS model, further solidifying the leading position of our proposed method. In summary, the PTE4TS model provides a new perspective and a powerful analytical tool for the field of time series analysis. We expect it to promote in-depth research in related areas and lay a solid foundation for building a more general and efficient AI framework for time series analysis. In the future, we will continue to explore and optimize Large Models in the time series, extending their application to a broader range of time series tasks, thereby advancing the development of the entire field.

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

# A    RELATED WORK

## A.1    LARGE MODEL IN TIME SERIES

Large Models have advantages in capturing long-term dependencies, handling high-dimensional data, and combating noise in the field of time series Zhao et al. (2023); Awais et al. (2023). They can be applied to a variety of specific scenarios including sleep analysis, weather prediction, and traffic flow forecasting. With the continuous development and innovation of technology, the potential applications of Large Models will be further unleashed, bringing more possibilities and value to time series analysis.

PromptCast Xue & Salim (2023) introduces a new paradigm for time series forecasting that leverages language models for prediction. In this paradigm, numerical inputs and outputs are transformed into prompts, and the forecasting task is constructed in a sentence-to-sentence manner. This allows for the direct application of language models to make predictions. The Frozen Pretrained Transformer (FPT) Zhou et al. (2023) improves time series tasks by utilizing Transformer Blocks from a pretrained language model, specifically GPT, for time series analysis. Additionally, the authors have discovered that the self-attention modules exhibit similarity to Principal Component Analysis (PCA) Wold et al. (1987) from both theoretical and empirical perspectives. This observation is instrumental in explaining how Transformers overcome disparities between domains, and it is a critical step toward understanding the universality of Transformers. TIME-LLM Jin et al. (2023a) achieves time series forecasting by repurposing LLMs. This method leverages the powerful pattern recognition and reasoning capabilities of LLMs on complex sequence data, aligning time series data with natural language to achieve accurate time series forecasting. LLM4TS Chang et al. (2023) enhances time series forecasting by leveraging pretrained LLMs. By combining methods of time series patching and time encoding, it improves the capability of LLMs to process time series data. TEST Sun et al. (2023) activates the capability of LLMs to process time series data by designing a time series embedding method suitable for LLMs. It first tokenizes the time series, then encodes the series through instance-level, feature-level, and text-prototype aligned contrastive learning methods. Next, it creates prompts to facilitate the LLM's understanding of the embedded information, ultimately enabling the execution of time series tasks. TEMPO Cao et al. (2023) introduces a generative pre-training Transformer model for representation learning and forecasting of time series. TEMPO decomposes time series into three components: trend, seasonality, and residuals, using locally weighted scatterplot smoothing. Each component is then mapped to a hidden space to construct the time series input embeddings for the generative pre-training Transformer. TEMPO uses a pool of prompts to guide the model's forecasting task, encoding temporal knowledge through the reuse of a set of learnable continuous vector representations. Most of these methods focus primarily on time series forecasting, with little or no attention to other applications within the time series domain. Although FPT is proposed to improve time series analysis tasks, it explores the application of LLMs in time series.

## A.2    PRE-TRAINING MODELS IN TIME SERIES

In addition to employing LLMs for time series analysis, the development of pre-training models and associated infrastructure based on time series data also holds significant potential and promise for application Jin et al. (2023b).

Voice2Series Yang et al. (2021) framework leverages the representational learning capabilities of pre-training voice processing models, adeptly converting voice data into univariate time signals for classification. CLUDA Ozyurt et al. (2022) stands as an innovative unsupervised domain adaptation framework for time series, employing custom and nearest-neighbor contrastive learning. Through contrastive learning, CLUDA aims to create a representational space where samples with semantic similarities are drawn closer together, while those with lesser affinity are distanced from each other, thereby facilitating the learning of contextual representations for multivariate time series across domains. STEP Shao et al. (2022) comprises a pre-training model and a spatio-temporal graph neural network (STGNN), where the pre-training model is focused on effectively capturing temporal patterns from long-term historical time series, generating segment-level representations to provide contextual information for short-term time series inputs to the STGNN, and enhancing the modeling of correlations among time series. MTSMAE Tang & Zhang (2022) is a self-supervised pretraining method developed from the foundational Masked Autoencoder (MAE) concept, specifically tai-

lored for multivariate time-series forecasting. The concept behind MTSMAE is straightforward: it segments the multivariate time series data into patches, then masks random patches from the input, and subsequently attempts to reconstruct the missing patches.The Patch embedding proposed by MTSMAE is instrumental in reducing memory usage, as it allows the model to handle longer sequences more efficiently. SimMTM Dong et al. (2023) extends the application of pre-training masked models to time series by uncovering the local structure of the manifolds. This model reconstructs masked time points through weighted aggregation of multiple neighbors outside the manifold, thereby improving the quality of sequence reconstruction. PatchTST Nie et al. (2022) is a Transformer-based model specifically designed for long-term time series forecasting. The model incorporates a patch mechanism to extract local semantic information, enabling each univariate sequence within the channel-independent architecture to learn its own attention map for accurate predictions. TSMixer Ekambaram et al. (2023) is a lightweight MLP-Mixer model tailored for multivariate time series forecasting. The model integrates two online reconciliation heads that fine-tune and enhance predictions by exploiting the hierarchical patch aggregation characteristics within the time series and the inter-channel correlations.

## B  PRE-TRAINING AND DOWNSTREAM TRAINING SETTINGS

### B.1  DATASET

Table 4: Long-term forecasting dataset descriptions.

| Dataset | Dim | Input Length | Prediction Length | Time Points | train/val/test | Information (Frequency) |
|---|---|---|---|---|---|---|
| ETTm1 ETTm2 | 7 | 336 | {96,192,336,720} | 69,680 | 6:2:2 | Electricity(15 mins) |
| ETTh1 ETTh2 | 7 | 336 | {96,192,336,720} | 17,420 | 6:2:2 | Electricity(15 mins) |
| Electricity | 321 | 336 | {96,192,336,720} | 26,304 | 7:1:2 | Electricity(Hourly) |
| Traffic | 862 | 336 | {96,192,336,720} | 17,544 | 7:1:2 | Transportation(Hourly) |
| Weather | 21 | 336 | {96,192,336,720} | 52,696 | 7:1:2 | Weather(10 mins) |
| Exchange | 8 | 336 | {96,192,336,720} | 7,588 | 7:1:2 | Exchange rate (Daily) |
| ILI | 7 | 104 | {24,36,48,60} | 966 | 7:1:2 | llness (Weekly) |

Timer has curated the Unified Time Series Dataset (UTSD) Liu et al. (2024), which encompasses seven domains and includes up to 1 billion time points, organized in a hierarchical four-volume structure to facilitate research and pre-training of large models in the time series field. The Unified Time Series Dataset (UTSD) is meticulously compiled from a combination of publicly accessible online data repositories and empirical data derived from real-world machine operations. All datasets are classified into seven distinct domains by their sources: Energy, Environment, Health, Internet of Things (IoT), Nature, Transportation, and Web, with diverse sampling frequencies. UTSD is structured with hierarchical capacities, namely UTSD-1G, UTSD-2G, UTSD-4G, and UTSD-12G, where each smaller dataset is a subset of the larger ones. A larger subset means greater data difficulty and diversity. For our pre-training tasks, we directly utilized data from the UTSD-12G.

In Table 4 and Table 5, we present detailed information about each dataset. These include Electricity, Weather, Traffic, ILI, as well as the ETT family comprising ETTh1, ETTh2, ETTm1, and ETTm2. Additionally, the framework is pre-trained on five standard time series anomaly detection datasets, specifically SMD, MSL, SMAP, SWaT, and PSM. Across these diverse datasets, a wide array of applications is covered, ranging from weather forecasting, traffic analysis, energy consumption, service monitoring, space & earth exploration, and water treatment technologies. Given the specificity of the downstream tasks associated with these datasets, our pre-training is strictly conducted on their respective training sets.

Table 5: Other dataset descriptions. The dataset size is organized in (Train,Validation,Test).

| Tasks | Dataset | Dim | Series Length | Dataset Size | Information (Frequency) |
|---|---|---|---|---|---|
| Imputation | ETTm1,ETTm2 | 7 | 96 | (34465,11521,11521) | Electricity(15 mins) |
| | ETTh1,ETTh2 | 7 | 96 | (8545,2881,2881) | Electricity (15 mins) |
| | Electricity | 321 | 96 | (18317,2633,5261) | Electricity(15 mins) |
| | Weather | 21 | 96 | (36792,5271,10540) | Weather(10 mins) |
| Classification (UEA) | EthanolConcentration | 3 | 1751 | (261,0,263) | Alcohol Industry |
| | FaceDetection | 144 | 62 | (5890,0,3524) | Face (250Hz) |
| | Handwriting | 3 | 152 | (150,0.850 | Handwriting |
| | Heartbeat | 61 | 405 | (204,0,205) | Heart Beat |
| | JapaneseVowels | 12 | 29 | (270,0,370) | Voice |
| | PEMS-SF | 963 | 144 | (267,0,173) | Transportation (Daily) |
| | SelfRegulationSCP1 | 6 | 896 | (268,0,293) | Health(256Hz) |
| | SelfRegulationSCP2 | 7 | 1152 | (200,0,180) | Health(256Hz) |
| | SpokenArabicDigits | 13 | 93 | (6599,0,2199) | Voice(11025Hz) |
| | UWaveGestureLibrary | 3 | 315 | (120,0,320) | Gesture |
| Anomaly Detection | SMD | 38 | 100 | (566724,141681,708420) | Server Machine |
| | MSL | 55 | 100 | (44653,11664,73729) | Spacecraft |
| | SMAP | 25 | 100 | (108146,27037,427617) | Spacecraft |
| | SWaT | 51 | 100 | (396000,99000,449919) | Infrastructure |
| | PSM | 25 | 100 | (105984,26497,87841) | Server Machine |

## B.2 PRE-TRAINING

Our PTE4TS pre-training framework is illustrated as shown in Figure 2, where we conduct pre-training across UTSD and performing data augmentation, specifically by downsampling the original training data to simulate different sampling frequencies.

We developed a 6-layer Encoder Transformer with attention layers(each providing a 256-dimensional representation of the data it processes and 16 heads). The architecture included position-wise feed-forward networks with an expansive 2048-dimensional inner state, allowing for rich internal representations to be formed during the learning process. Each iteration of our training provides the model with mini-batches composed of 1024 randomly selected small batch data, with each sequence capable of accepting up to 1440 input tokens. Optimization was carried out using the Adam scheme Kingma & Ba (2014), with peak learning rates capping at 0.0001. Diverging from the sine-based position embedding proposed in the foundational literature, we implemented learned positional embedding, solidifying their contributions to the model's understanding of sequence by locking them during fine-tuning stages. By freezing these embedding in downstream tasks, we ensured their gradients remained untouched, preserving their intrinsic values for consistent performance.

In the pre-training task, we did not set a fixed random number seed and conducted experiments on four NVIDIA V100 16GB GPUs. We selected an exceptionally high mask ratio of 75%. Due to the absence of a fixed random seed, this can largely guarantee that the data inputted into the model in each round of the pre-training phase is different.

## B.3 DOWNSTREAM TASKS

For our downstream tasks, we've tailored the Projection module into a specialized output layer to cater to diverse tasks while maintaining the rest of the architecture intact. The model will then undergo fine-tuning on specific datasets. Unless otherwise specified, we will adopt the hyperparameter configuration from the pre-training phase. Our optimization approach for downstream tasks will align with the protocols set by TimesNet Wu et al. (2022).

We did not set a random seed and each experiment was repeated six times to obtain the mean value; these were conducted on a single NVIDIA V100 16GB GPU. We employed early stopping, that

918 is, training was halted if there was no improvement in performance on the Validation set for five
919 consecutive epochs.
920
921 All experiments were implemented in PyTorch Paszke et al. (2019). For pre-training, long-term
922 forecasting and imputations, we adopt the mean square error (MSE) and mean absolute error (MAE)
923 for the metrics. For anomaly detection, we adopt the F1-score, which is the harmonic mean of
924 Precision and Rrecall. Apart from utilizing cross-entropy as the loss function for classification tasks,
925 MSE was employed as the loss function for all other tasks.

## C  MODEL ANALYSIS

### C.1  ATTENTION TYPE

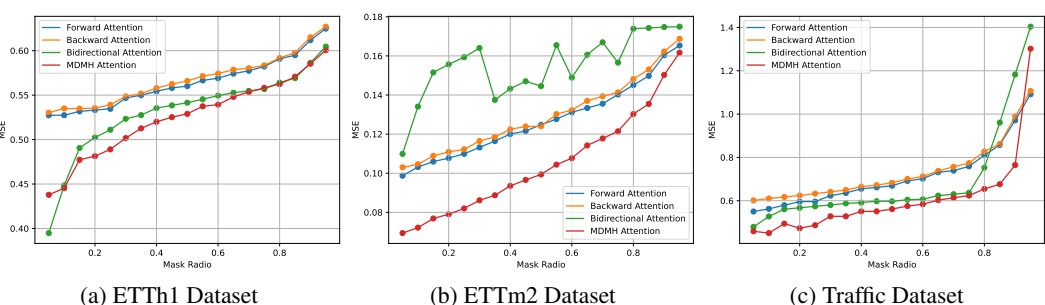

(a) ETTh1 Dataset        (b) ETTm2 Dataset        (c) Traffic Dataset

Figure 5: The MSE results in different long-term forecasting datasets after masking the input at different scales on Patch-Level and passing through different types of Attention.

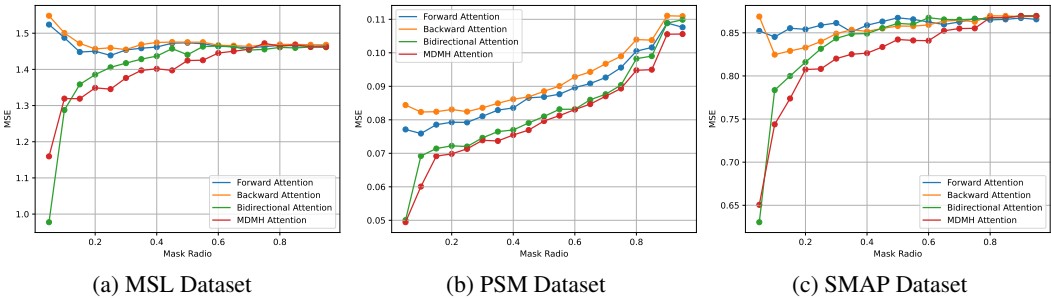

(a) MSL Dataset        (b) PSM Dataset        (c) SMAP Dataset

Figure 6: The MSE results in different anomaly detection datasets after masking the input at different scales on Patch-Level and passing through different types of Attention.

In vanilla Attention, the positional relationship amongst elements is not directly accounted for, resulting in the architecture's inability to effectively capture the critical information of element sequence within the input series. This issue is somewhat alleviated through the use of positional embdedding. However, such sequential information is vital for the analysis of time series data, which often exhibits a pronounced sequential dependency; that is, the occurrence of one event might be directly influenced by a preceding event. The employment of lower or upper triangular masks can serve as a more effective approach to handling positional embdedding information. Essentially, the triangular mask undermines the permutation invariance characteristic of the Transformer model, introducing a definitive left-to-right sequence order. Specifically, in the presence of a triangular mask, subsequent tokens in the sequence can only attend to preceding tokens, not to following ones, thereby inherently incorporating positional information.

Furthermore, the utilization of a full-rank weight matrix implies that the network can capture a broader array of input data features, making full use of model parameters to provide a richer representational capacity. In the absence of additional restrictive conditions, a full-rank matrix may exhibit a strong expressive ability, as it can represent a wider range of transformations. Within the context of attention models, expressive capability generally refers to the model's capacity to capture

and utilize information, as well as its ability to model complex relationships. Although the property of full-rank does not inherently guarantee high expressive capability, it bestows upon the model the potential for such capacity.

Our discussion specifically examines the role of Attention mechanisms of various orientations in a range of time series tasks. Figure 5 presents the MSE in pre-training tasks after applying different types of Attention mechanisms on three long-term forecasting datasets. It is evident that although BiDirectional Attention is not full-rank, its consideration of bidirectional interactions renders it superior to UniDirectional Attention to a certain extent. However, in the ETTm2 dataset, BiDirectional Attention underperforms compared to UniDirectional Attention, which, to some extent, demonstrates the advantages of full-rank Attention.

Table 6: The results of fine-tuning and no pre-training using different types of Attention on long-term forecasting datasets

| Methods | | MDMH Attention | | | | Bidirectional Attention | | | | Forward Attention | | | | Backward Attention | | | |
|---|---|---|---|---|---|---|---|---|---|---|---|---|---|---|---|---|---|
| | | Fine-tuning | | Supervised | | Fine-tuning | | Supervised | | Fine-tuning | | Supervised | | Fine-tuning | | Supervised | |
| Metric | | MSE | MAE | MSE | MAE | MSE | MAE | MSE | MAE | MSE | MAE | MSE | MAE | MSE | MAE | MSE | MAE |
| ETTh2 | 96 | 0.271 | 0.329 | 0.280 | 0.336 | 0.276 | 0.334 | 0.281 | 0.339 | 0.282 | 0.339 | 0.288 | 0.343 | 0.279 | 0.335 | 0.282 | 0.339 |
| | 192 | 0.337 | 0.372 | 0.351 | 0.384 | 0.346 | 0.379 | 0.350 | 0.385 | 0.346 | 0.381 | 0.355 | 0.386 | 0.344 | 0.378 | 0.347 | 0.382 |
| | 336 | 0.331 | 0.377 | 0.343 | 0.389 | 0.331 | 0.380 | 0.337 | 0.386 | 0.336 | 0.385 | 0.337 | 0.388 | 0.340 | 0.384 | 0.342 | 0.389 |
| | 720 | 0.383 | 0.420 | 0.393 | 0.431 | 0.377 | 0.419 | 0.384 | 0.425 | 0.382 | 0.421 | 0.391 | 0.431 | 0.390 | 0.427 | 0.393 | 0.430 |
| | Avg | **0.331** | **0.375** | 0.342 | 0.385 | 0.333 | 0.378 | 0.338 | 0.384 | 0.337 | 0.382 | 0.343 | 0.387 | 0.338 | 0.381 | 0.341 | 0.385 |
| ETTm1 | 96 | 0.277 | 0.331 | 0.286 | 0.338 | 0.280 | 0.333 | 0.286 | 0.339 | 0.282 | 0.336 | 0.285 | 0.338 | 0.282 | 0.336 | 0.288 | 0.339 |
| | 192 | 0.318 | 0.357 | 0.325 | 0.363 | 0.321 | 0.359 | 0.330 | 0.367 | 0.324 | 0.364 | 0.329 | 0.367 | 0.322 | 0.362 | 0.326 | 0.363 |
| | 336 | 0.352 | 0.378 | 0.360 | 0.386 | 0.354 | 0.381 | 0.366 | 0.391 | 0.358 | 0.384 | 0.363 | 0.390 | 0.357 | 0.384 | 0.360 | 0.385 |
| | 720 | 0.397 | 0.408 | 0.407 | 0.416 | 0.403 | 0.412 | 0.411 | 0.417 | 0.406 | 0.412 | 0.410 | 0.419 | 0.404 | 0.412 | 0.410 | 0.417 |
| | Avg | **0.336** | **0.369** | 0.345 | 0.376 | 0.340 | 0.371 | 0.348 | 0.379 | 0.343 | 0.374 | 0.347 | 0.379 | 0.341 | 0.374 | 0.346 | 0.376 |
| Exchange | 96 | 0.082 | 0.201 | 0.090 | 0.210 | 0.082 | 0.203 | 0.091 | 0.211 | 0.085 | 0.206 | 0.088 | 0.207 | 0.087 | 0.205 | 0.089 | 0.209 |
| | 192 | 0.179 | 0.300 | 0.192 | 0.313 | 0.183 | 0.305 | 0.200 | 0.319 | 0.187 | 0.309 | 0.197 | 0.318 | 0.185 | 0.306 | 0.194 | 0.314 |
| | 336 | 0.351 | 0.427 | 0.372 | 0.442 | 0.349 | 0.428 | 0.371 | 0.444 | 0.353 | 0.432 | 0.359 | 0.436 | 0.351 | 0.428 | 0.369 | 0.441 |
| | 720 | 0.854 | 0.683 | 0.916 | 0.708 | 0.894 | 0.707 | 0.880 | 0.694 | 0.870 | 0.688 | 0.893 | 0.700 | 0.898 | 0.709 | 0.955 | 0.724 |
| | Avg | **0.367** | **0.403** | 0.393 | 0.418 | 0.377 | 0.411 | 0.386 | 0.417 | 0.374 | 0.409 | 0.384 | 0.415 | 0.380 | 0.412 | 0.402 | 0.422 |
| Weather | 96 | 0.148 | 0.197 | 0.154 | 0.204 | 0.151 | 0.198 | 0.158 | 0.207 | 0.153 | 0.201 | 0.157 | 0.204 | 0.151 | 0.200 | 0.157 | 0.207 |
| | 192 | 0.191 | 0.235 | 0.197 | 0.243 | 0.194 | 0.239 | 0.198 | 0.243 | 0.195 | 0.24 | 0.198 | 0.244 | 0.195 | 0.240 | 0.200 | 0.246 |
| | 336 | 0.241 | 0.275 | 0.248 | 0.282 | 0.243 | 0.277 | 0.249 | 0.282 | 0.245 | 0.279 | 0.248 | 0.282 | 0.246 | 0.280 | 0.249 | 0.284 |
| | 720 | 0.313 | 0.325 | 0.320 | 0.333 | 0.314 | 0.328 | 0.321 | 0.335 | 0.315 | 0.329 | 0.319 | 0.333 | 0.317 | 0.332 | 0.321 | 0.335 |
| | Avg | **0.222** | **0.257** | 0.230 | 0.266 | 0.226 | 0.261 | 0.232 | 0.267 | 0.227 | 0.262 | 0.231 | 0.266 | 0.227 | 0.263 | 0.232 | 0.268 |
| ECL | 96 | 0.128 | 0.224 | 0.136 | 0.231 | 0.131 | 0.227 | 0.136 | 0.232 | 0.133 | 0.228 | 0.138 | 0.234 | 0.135 | 0.230 | 0.138 | 0.234 |
| | 192 | 0.143 | 0.234 | 0.151 | 0.245 | 0.146 | 0.241 | 0.154 | 0.248 | 0.149 | 0.242 | 0.152 | 0.246 | 0.149 | 0.243 | 0.153 | 0.247 |
| | 336 | 0.159 | 0.253 | 0.168 | 0.263 | 0.162 | 0.256 | 0.168 | 0.262 | 0.166 | 0.261 | 0.170 | 0.264 | 0.165 | 0.260 | 0.169 | 0.263 |
| | 720 | 0.199 | 0.286 | 0.208 | 0.296 | 0.202 | 0.289 | 0.207 | 0.295 | 0.204 | 0.293 | 0.207 | 0.295 | 0.204 | 0.292 | 0.208 | 0.296 |
| | Avg | **0.157** | **0.249** | 0.166 | 0.259 | 0.160 | 0.253 | 0.166 | 0.259 | 0.163 | 0.256 | 0.167 | 0.260 | 0.163 | 0.256 | 0.167 | 0.260 |
| Traffic | 96 | 0.383 | 0.260 | 0.390 | 0.268 | 0.386 | 0.265 | 0.394 | 0.272 | 0.396 | 0.271 | 0.398 | 0.275 | 0.390 | 0.268 | 0.393 | 0.273 |
| | 192 | 0.397 | 0.266 | 0.406 | 0.275 | 0.401 | 0.272 | 0.409 | 0.278 | 0.407 | 0.274 | 0.411 | 0.279 | 0.407 | 0.274 | 0.409 | 0.278 |
| | 336 | 0.408 | 0.268 | 0.416 | 0.280 | 0.413 | 0.276 | 0.419 | 0.282 | 0.417 | 0.282 | 0.423 | 0.285 | 0.417 | 0.282 | 0.421 | 0.287 |
| | 720 | 0.437 | 0.289 | 0.445 | 0.297 | 0.441 | 0.294 | 0.448 | 0.300 | 0.444 | 0.296 | 0.449 | 0.301 | 0.445 | 0.297 | 0.450 | 0.302 |
| | Avg | **0.406** | **0.271** | 0.414 | 0.280 | 0.410 | 0.277 | 0.418 | 0.283 | 0.416 | 0.281 | 0.420 | 0.285 | 0.415 | 0.280 | 0.418 | 0.285 |

It is noteworthy that the Multi-Directional Multi-Head (MDMH) Attention we propose not only accounts for bidirectional information input but also possesses full-rank characteristics. Consequently, it achieves the best results in recovery tasks across all datasets. As shown in Talbe 6, we conducted a detailed analysis of the impact of different directional Attention mechanisms on forecasting tasks and fine-tuning for forecasts. It is apparent that MDMH continues to achieve state-of-the-art (SOTA) results across all datasets. Simultaneously, it is observed that there seems to be little difference between Forward Attention (FA) and Backward Attention (BA), suggesting that causality is not particularly crucial for predicting time series.

Furthermore, Figure 6 and Table 7 display experimental results of different directional Attention mechanisms on various tasks within anomaly detection datasets. Figure 7 demonstrates the classification accuracy of different directional Attention in datasets focused on classification. In re-

construction tasks, MDMH consistently shows superior performance. In anomaly detection tasks, MDMH exhibits the best performance in multiple datasets, which can be attributed to the fact that although MDMH incorporates information in two directions, it lacks the interaction between these two directions.

Table 7: The results of fine-tuning and no pre-training using different types of Attention on anomaly detection datasets

| Dataset | Metric | MDMH Attention | | Bidirectional Attention | | Forward Attention | | Backward Attention | |
|---|---|---|---|---|---|---|---|---|---|
| | | Fine-tuning | Supervised | Fine-tuning | Supervised | Fine-tuning | Supervised | Fine-tuning | Supervised |
| MSL | Accuracy | 96.32 | 96.11 | 96.55 | 96.34 | 96.22 | 95.75 | 95.91 | 95.21 |
| | Precision | 89.33 | 88.86 | 90.01 | 89.47 | 89.31 | 88.67 | 88.56 | 86.86 |
| | Recall | 74.12 | 72.18 | 75.23 | 74.05 | 71.89 | 68.42 | 70.34 | 64.36 |
| | F-score | 81.00 | 79.66 | **81.99** | 81.03 | 79.65 | 77.24 | 78.40 | 73.93 |
| PSM | Accuracy | 97.67 | 97.60 | 97.54 | 97.49 | 97.51 | 97.41 | 97.54 | 97.48 |
| | Precision | 98.56 | 98.50 | 98.40 | 98.43 | 98.43 | 98.33 | 98.38 | 98.48 |
| | Recall | 92.95 | 92.74 | 92.66 | 92.41 | 92.52 | 92.23 | 92.67 | 92.36 |
| | F-score | **95.67** | 95.53 | 95.44 | 95.33 | 95.38 | 95.18 | 95.44 | 95.32 |
| SMAP | Accuracy | 93.54 | 93.21 | 95.02 | 94.25 | 93.87 | 93.47 | 93.89 | 93.48 |
| | Precision | 90.43 | 89.91 | 92.26 | 91.24 | 90.45 | 90.28 | 90.67 | 90.33 |
| | Recall | 55.37 | 52.85 | 62.14 | 60.91 | 55.25 | 54.87 | 55.26 | 54.94 |
| | F-score | 68.68 | 66.57 | **74.26** | 73.05 | 68.59 | 68.25 | 69.09 | 68.32 |
| SMD | Accuracy | 98.87 | 98.51 | 98.50 | 98.48 | 98.48 | 98.43 | 98.50 | 98.49 |
| | Precision | 88.69 | 88.27 | 88.24 | 88.20 | 88.24 | 88.18 | 88.36 | 88.23 |
| | Recall | 74.43 | 73.84 | 73.60 | 73.31 | 73.20 | 73.12 | 73.55 | 73.49 |
| | F-score | **80.93** | 80.47 | 80.26 | 80.07 | 80.02 | 79.98 | 80.28 | 80.19 |
| SWAT | Accuracy | 98.48 | 98.13 | 98.45 | 98.11 | 98.12 | 98.11 | 98.14 | 98.12 |
| | Precision | 92.84 | 92.19 | 92.84 | 92.12 | 92.20 | 92.16 | 92.23 | 92.18 |
| | Recall | 93.01 | 92.38 | 92.98 | 92.36 | 93.29 | 92.26 | 92.39 | 92.34 |
| | F-score | **92.97** | 92.27 | 92.96 | 92.24 | 92.74 | 92.21 | 92.31 | 92.26 |

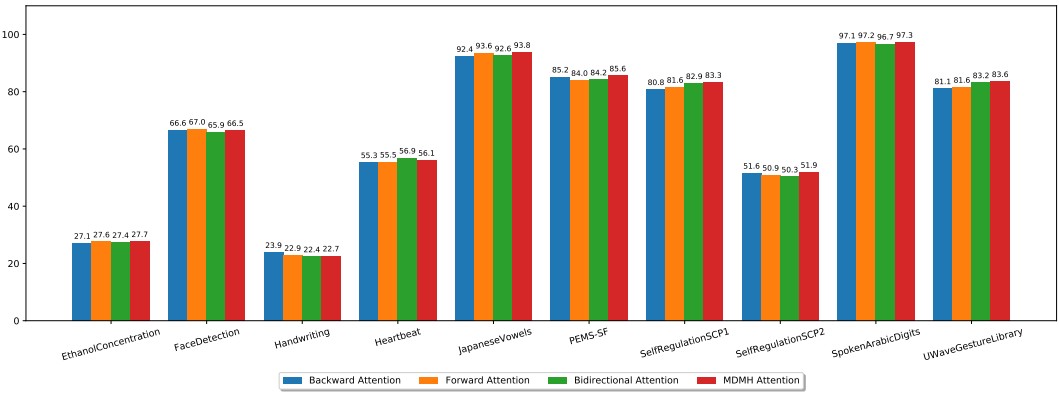

Figure 7: The results of different classification datasets passing through different types of Attention.

## C.2 MASK TYPE

The reconstruction task at the patch-level involves masking the input on a patch basis, rendering large sequences invisible and creating a more challenging recovery scenario. This allows the model to better learn local information. However, in situations with an exceedingly high mask ratio, too much information may be lost, causing the model to learn only repetitive patterns rather than global trends or other relevant data.

In contrast, point-level reconstruction tasks mask inputs on a point basis, which helps preserve the trend information of sequences. Although subsequent patching can enhance the model's ability to learn local features, distinguishing whether local anomalies are due to masking or noise points

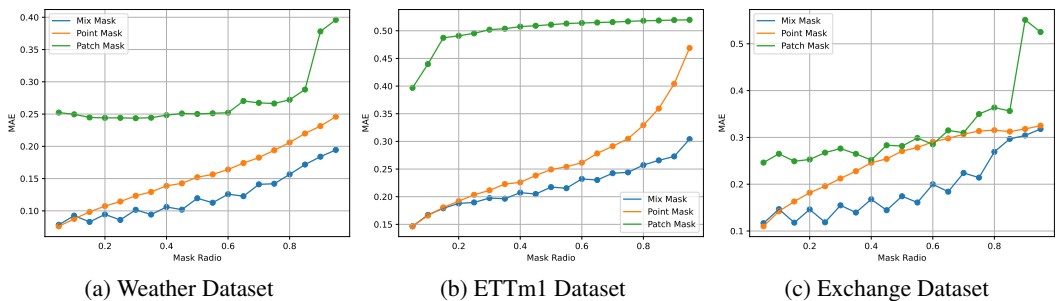

(a) Weather Dataset        (b) ETTm1 Dataset        (c) Exchange Dataset

Figure 8: After using different masking methods to obscure the input at various ratios, the MAE results in different long-torm forecasting datasets are obtained following Bidirectional Attention.

becomes challenging for the model. With a very high mask ratio, the task remains difficult, yet lost information is relatively easier to recover from neighboring points.

The mix mask approach that incorporates both scrambling within patches and mean replacement, establishing a patch-level reconstruction task. This approach allows models to learn the order of local information to some extent, while point-level masking ensures that global trends and other information are not lost.

Figure 8 demonstrates the results of employing different masking methods in reconstruction tasks across three long-term prediction datasets. It clearly shows that, despite integrating some aspects of patch-level masking, the mixed mask approach still outperforms point-level masking. This highlights that the mixed strategy enables models to learn global information without neglecting local details and can, to a certain extent, learn sequential information.

# D    FULL RESULTS

## D.1    FULL RESULTS OF LONG-TERM TIME-SERIES FORECASTING

## D.2    FULL RESULTS OF IMPUTATION

## D.3    FULL RESULTS OF ANOMALY DETECTION

Table 8: Full Results of Long-term Time-series Forecasting

| Methods | | PTE4TS | | FPT | | DLinear | | PatchTST | | TimesNet | | FEDformer | | Autoformer | | Stationary | | ETSformer | | LightTS | |
|---|---|---|---|---|---|---|---|---|---|---|---|---|---|---|---|---|---|---|---|---|---|
| Metric | | MSE | MAE | MSE | MAE | MSE | MAE | MSE | MAE | MSE | MAE | MSE | MAE | MSE | MAE | MSE | MAE | MSE | MAE | MSE | MAE |
| Weather | 96 | 0.139 | 0.187 | 0.162 | 0.212 | 0.176 | 0.237 | 0.149 | 0.198 | 0.172 | 0.220 | 0.217 | 0.296 | 0.266 | 0.336 | 0.173 | 0.223 | 0.197 | 0.281 | 0.182 | 0.242 |
| | 192 | 0.182 | 0.230 | 0.204 | 0.248 | 0.220 | 0.282 | 0.194 | 0.241 | 0.219 | 0.261 | 0.276 | 0.336 | 0.307 | 0.367 | 0.245 | 0.285 | 0.237 | 0.312 | 0.227 | 0.287 |
| | 336 | 0.232 | 0.268 | 0.254 | 0.286 | 0.265 | 0.319 | 0.245 | 0.282 | 0.280 | 0.306 | 0.339 | 0.380 | 0.359 | 0.395 | 0.321 | 0.338 | 0.298 | 0.353 | 0.282 | 0.334 |
| | 720 | 0.306 | 0.321 | 0.326 | 0.337 | 0.333 | 0.362 | 0.314 | 0.334 | 0.365 | 0.359 | 0.403 | 0.428 | 0.419 | 0.428 | 0.414 | 0.410 | 0.352 | 0.288 | 0.352 | 0.386 |
| | Avg | 0.215 | 0.251 | 0.237 | 0.270 | 0.248 | 0.300 | 0.225 | 0.264 | 0.259 | 0.287 | 0.309 | 0.360 | 0.338 | 0.382 | 0.288 | 0.314 | 0.271 | 0.334 | 0.261 | 0.312 |
| ETTh1 | 96 | 0.358 | 0.381 | 0.376 | 0.397 | 0.375 | 0.399 | 0.370 | 0.399 | 0.384 | 0.402 | 0.376 | 0.419 | 0.449 | 0.459 | 0.513 | 0.491 | 0.494 | 0.479 | 0.424 | 0.432 |
| | 192 | 0.405 | 0.413 | 0.416 | 0.418 | 0.405 | 0.416 | 0.413 | 0.421 | 0.436 | 0.429 | 0.420 | 0.448 | 0.500 | 0.482 | 0.534 | 0.504 | 0.538 | 0.504 | 0.475 | 0.462 |
| | 336 | 0.413 | 0.425 | 0.442 | 0.433 | 0.439 | 0.443 | 0.422 | 0.436 | 0.491 | 0.469 | 0.459 | 0.465 | 0.521 | 0.496 | 0.588 | 0.535 | 0.574 | 0.521 | 0.518 | 0.488 |
| | 720 | 0.438 | 0.453 | 0.477 | 0.456 | 0.472 | 0.490 | 0.447 | 0.466 | 0.521 | 0.500 | 0.506 | 0.507 | 0.514 | 0.512 | 0.643 | 0.616 | 0.562 | 0.535 | 0.547 | 0.533 |
| | Avg | 0.404 | 0.418 | 0.427 | 0.426 | 0.422 | 0.437 | 0.413 | 0.430 | 0.458 | 0.450 | 0.440 | 0.460 | 0.496 | 0.487 | 0.570 | 0.537 | 0.542 | 0.510 | 0.491 | 0.479 |
| ETTh2 | 96 | 0.262 | 0.324 | 0.285 | 0.342 | 0.289 | 0.353 | 0.274 | 0.336 | 0.340 | 0.374 | 0.358 | 0.397 | 0.346 | 0.388 | 0.476 | 0.458 | 0.340 | 0.391 | 0.397 | 0.437 |
| | 192 | 0.330 | 0.369 | 0.354 | 0.389 | 0.383 | 0.418 | 0.339 | 0.379 | 0.402 | 0.414 | 0.429 | 0.439 | 0.456 | 0.452 | 0.512 | 0.493 | 0.430 | 0.439 | 0.520 | 0.504 |
| | 336 | 0.321 | 0.368 | 0.373 | 0.407 | 0.448 | 0.465 | 0.329 | 0.380 | 0.452 | 0.452 | 0.496 | 0.487 | 0.482 | 0.486 | 0.552 | 0.551 | 0.485 | 0.479 | 0.626 | 0.559 |
| | 720 | 0.373 | 0.413 | 0.406 | 0.441 | 0.605 | 0.551 | 0.379 | 0.422 | 0.462 | 0.468 | 0.463 | 0.474 | 0.515 | 0.511 | 0.562 | 0.560 | 0.500 | 0.497 | 0.863 | 0.672 |
| | Avg | 0.322 | 0.368 | 0.354 | 0.394 | 0.431 | 0.446 | 0.330 | 0.379 | 0.414 | 0.427 | 0.437 | 0.449 | 0.450 | 0.459 | 0.526 | 0.516 | 0.439 | 0.452 | 0.602 | 0.543 |
| ETTm1 | 96 | 0.270 | 0.325 | 0.292 | 0.346 | 0.299 | 0.343 | 0.290 | 0.342 | 0.338 | 0.375 | 0.379 | 0.419 | 0.505 | 0.475 | 0.386 | 0.398 | 0.375 | 0.398 | 0.374 | 0.400 |
| | 192 | 0.313 | 0.348 | 0.332 | 0.372 | 0.335 | 0.365 | 0.332 | 0.369 | 0.374 | 0.387 | 0.426 | 0.441 | 0.553 | 0.496 | 0.459 | 0.444 | 0.408 | 0.410 | 0.400 | 0.407 |
| | 336 | 0.343 | 0.368 | 0.366 | 0.394 | 0.369 | 0.386 | 0.366 | 0.392 | 0.410 | 0.411 | 0.445 | 0.459 | 0.621 | 0.537 | 0.495 | 0.464 | 0.435 | 0.428 | 0.438 | 0.438 |
| | 720 | 0.388 | 0.400 | 0.417 | 0.421 | 0.425 | 0.421 | 0.416 | 0.420 | 0.478 | 0.450 | 0.543 | 0.490 | 0.671 | 0.561 | 0.585 | 0.516 | 0.499 | 0.462 | 0.527 | 0.502 |
| | Avg | 0.328 | 0.360 | 0.352 | 0.383 | 0.357 | 0.378 | 0.351 | 0.380 | 0.400 | 0.406 | 0.448 | 0.452 | 0.588 | 0.517 | 0.481 | 0.456 | 0.429 | 0.425 | 0.435 | 0.437 |
| ETTm2 | 96 | 0.155 | 0.245 | 0.173 | 0.262 | 0.167 | 0.269 | 0.165 | 0.255 | 0.187 | 0.267 | 0.203 | 0.287 | 0.255 | 0.339 | 0.192 | 0.274 | 0.189 | 0.280 | 0.209 | 0.308 |
| | 192 | 0.211 | 0.279 | 0.229 | 0.301 | 0.224 | 0.303 | 0.220 | 0.292 | 0.249 | 0.309 | 0.269 | 0.328 | 0.281 | 0.340 | 0.280 | 0.339 | 0.253 | 0.319 | 0.311 | 0.382 |
| | 336 | 0.263 | 0.315 | 0.286 | 0.341 | 0.281 | 0.342 | 0.274 | 0.329 | 0.321 | 0.351 | 0.325 | 0.366 | 0.339 | 0.372 | 0.334 | 0.361 | 0.314 | 0.357 | 0.442 | 0.466 |
| | 720 | 0.353 | 0.374 | 0.378 | 0.401 | 0.397 | 0.421 | 0.362 | 0.385 | 0.408 | 0.403 | 0.421 | 0.415 | 0.433 | 0.432 | 0.417 | 0.413 | 0.414 | 0.413 | 0.675 | 0.587 |
| | Avg | 0.246 | 0.303 | 0.266 | 0.326 | 0.267 | 0.333 | 0.255 | 0.315 | 0.291 | 0.333 | 0.305 | 0.349 | 0.327 | 0.371 | 0.306 | 0.347 | 0.293 | 0.342 | 0.409 | 0.436 |
| ILI | 24 | 1.207 | 0.689 | 2.063 | 0.881 | 2.215 | 1.081 | 1.319 | 0.754 | 2.317 | 0.934 | 3.228 | 1.260 | 3.483 | 1.287 | 2.294 | 0.945 | 2.527 | 1.020 | 8.313 | 2.144 |
| | 36 | 1.313 | 0.761 | 1.868 | 0.892 | 1.963 | 0.963 | 1.430 | 0.834 | 1.972 | 0.920 | 2.679 | 1.080 | 3.103 | 1.148 | 1.825 | 0.848 | 2.615 | 1.007 | 6.631 | 1.902 |
| | 48 | 1.427 | 0.725 | 1.790 | 0.884 | 2.130 | 1.024 | 1.553 | 0.815 | 2.238 | 0.940 | 2.622 | 1.078 | 2.669 | 1.085 | 2.010 | 0.900 | 2.359 | 0.972 | 7.299 | 1.982 |
| | 60 | 1.351 | 0.710 | 1.979 | 0.957 | 2.368 | 1.096 | 1.470 | 0.788 | 2.027 | 0.928 | 2.857 | 1.157 | 2.770 | 1.125 | 2.178 | 0.963 | 2.487 | 1.016 | 7.283 | 1.985 |
| | Avg | 1.324 | 0.721 | 1.925 | 0.903 | 2.169 | 1.041 | 1.443 | 0.797 | 2.139 | 0.931 | 2.847 | 1.144 | 3.006 | 1.161 | 2.077 | 0.914 | 2.497 | 1.004 | 7.382 | 2.003 |
| ECL | 96 | 0.120 | 0.213 | 0.139 | 0.238 | 0.140 | 0.237 | 0.129 | 0.222 | 0.168 | 0.272 | 0.193 | 0.308 | 0.201 | 0.317 | 0.169 | 0.273 | 0.187 | 0.304 | 0.207 | 0.307 |
| | 192 | 0.135 | 0.225 | 0.153 | 0.251 | 0.153 | 0.249 | 0.157 | 0.240 | 0.184 | 0.289 | 0.201 | 0.315 | 0.222 | 0.334 | 0.182 | 0.286 | 0.199 | 0.315 | 0.213 | 0.316 |
| | 336 | 0.145 | 0.244 | 0.169 | 0.266 | 0.169 | 0.267 | 0.163 | 0.259 | 0.198 | 0.300 | 0.214 | 0.329 | 0.231 | 0.338 | 0.200 | 0.304 | 0.212 | 0.329 | 0.230 | 0.333 |
| | 720 | 0.181 | 0.270 | 0.206 | 0.297 | 0.203 | 0.301 | 0.197 | 0.290 | 0.220 | 0.320 | 0.246 | 0.355 | 0.254 | 0.361 | 0.222 | 0.321 | 0.233 | 0.345 | 0.265 | 0.360 |
| | Avg | 0.145 | 0.238 | 0.167 | 0.263 | 0.166 | 0.263 | 0.161 | 0.252 | 0.192 | 0.295 | 0.214 | 0.327 | 0.227 | 0.338 | 0.193 | 0.296 | 0.208 | 0.323 | 0.229 | 0.329 |
| Traffic | 96 | 0.354 | 0.244 | 0.388 | 0.282 | 0.410 | 0.282 | 0.360 | 0.249 | 0.593 | 0.321 | 0.587 | 0.366 | 0.613 | 0.388 | 0.612 | 0.338 | 0.607 | 0.392 | 0.615 | 0.391 |
| | 192 | 0.374 | 0.251 | 0.407 | 0.290 | 0.423 | 0.287 | 0.379 | 0.256 | 0.617 | 0.336 | 0.604 | 0.373 | 0.616 | 0.382 | 0.613 | 0.340 | 0.621 | 0.399 | 0.601 | 0.382 |
| | 336 | 0.383 | 0.254 | 0.412 | 0.294 | 0.436 | 0.296 | 0.392 | 0.264 | 0.629 | 0.336 | 0.621 | 0.383 | 0.622 | 0.337 | 0.618 | 0.328 | 0.622 | 0.396 | 0.613 | 0.386 |
| | 720 | 0.416 | 0.271 | 0.450 | 0.312 | 0.466 | 0.315 | 0.432 | 0.286 | 0.640 | 0.350 | 0.626 | 0.382 | 0.660 | 0.408 | 0.653 | 0.355 | 0.632 | 0.396 | 0.658 | 0.407 |
| | Avg | 0.382 | 0.255 | 0.414 | 0.294 | 0.433 | 0.295 | 0.390 | 0.263 | 0.620 | 0.336 | 0.610 | 0.376 | 0.628 | 0.379 | 0.624 | 0.340 | 0.621 | 0.396 | 0.622 | 0.392 |

Table 9: Full Results of Imputation

| Methods | | PTE4TS | | FPT | | DLinear | | PatchTST | | TimesNet | | FEDformer | | Autoformer | | Stationary | | ETSformer | | LightTS | |
|---|---|---|---|---|---|---|---|---|---|---|---|---|---|---|---|---|---|---|---|---|---|
| Metric | | MSE | MAE | MSE | MAE | MSE | MAE | MSE | MAE | MSE | MAE | MSE | MAE | MSE | MAE | MSE | MAE | MSE | MAE | MSE | MAE |
| ETTm1 | 12.5% | 0.011 | 0.075 | 0.017 | 0.085 | 0.023 | 0.101 | 0.041 | 0.130 | 0.096 | 0.229 | 0.093 | 0.206 | 0.080 | 0.193 | 0.052 | 0.166 | 0.032 | 0.119 | 0.046 | 0.144 |
| | 25% | 0.015 | 0.088 | 0.022 | 0.096 | 0.023 | 0.101 | 0.044 | 0.135 | 0.096 | 0.229 | 0.093 | 0.206 | 0.080 | 0.193 | 0.052 | 0.166 | 0.032 | 0.119 | 0.046 | 0.144 |
| | 37.5% | 0.025 | 0.104 | 0.029 | 0.111 | 0.029 | 0.111 | 0.049 | 0.143 | 0.133 | 0.271 | 0.113 | 0.231 | 0.103 | 0.219 | 0.069 | 0.191 | 0.039 | 0.131 | 0.057 | 0.161 |
| | 50% | 0.034 | 0.115 | 0.040 | 0.128 | 0.036 | 0.124 | 0.055 | 0.151 | 0.186 | 0.323 | 0.134 | 0.255 | 0.132 | 0.248 | 0.089 | 0.218 | 0.047 | 0.145 | 0.067 | 0.174 |
| | Avg | 0.021 | 0.096 | 0.028 | 0.105 | 0.027 | 0.107 | 0.047 | 0.140 | 0.120 | 0.253 | 0.104 | 0.218 | 0.093 | 0.206 | 0.062 | 0.177 | 0.036 | 0.126 | 0.051 | 0.150 |
| ETTm2 | 12.5% | 0.010 | 0.062 | 0.017 | 0.076 | 0.018 | 0.080 | 0.026 | 0.094 | 0.108 | 0.239 | 0.034 | 0.127 | 0.062 | 0.166 | 0.056 | 0.159 | 0.021 | 0.088 | 0.023 | 0.092 |
| | 25% | 0.016 | 0.073 | 0.020 | 0.080 | 0.020 | 0.085 | 0.028 | 0.099 | 0.164 | 0.294 | 0.042 | 0.143 | 0.085 | 0.196 | 0.080 | 0.195 | 0.024 | 0.096 | 0.026 | 0.101 |
| | 37.5% | 0.015 | 0.079 | 0.022 | 0.087 | 0.023 | 0.091 | 0.030 | 0.104 | 0.237 | 0.356 | 0.051 | 0.159 | 0.106 | 0.222 | 0.110 | 0.231 | 0.027 | 0.103 | 0.030 | 0.108 |
| | 50% | 0.019 | 0.084 | 0.025 | 0.095 | 0.026 | 0.098 | 0.034 | 0.110 | 0.323 | 0.421 | 0.059 | 0.174 | 0.131 | 0.247 | 0.156 | 0.276 | 0.030 | 0.108 | 0.035 | 0.119 |
| | Avg | 0.015 | 0.075 | 0.021 | 0.084 | 0.022 | 0.088 | 0.029 | 0.102 | 0.208 | 0.327 | 0.046 | 0.151 | 0.096 | 0.208 | 0.101 | 0.215 | 0.026 | 0.099 | 0.029 | 0.105 |
| ETTh1 | 12.5% | 0.036 | 0.132 | 0.043 | 0.140 | 0.057 | 0.159 | 0.093 | 0.201 | 0.126 | 0.263 | 0.240 | 0.345 | 0.151 | 0.267 | 0.070 | 0.190 | 0.060 | 0.165 | 0.074 | 0.182 |
| | 25% | 0.046 | 0.148 | 0.054 | 0.156 | 0.069 | 0.178 | 0.107 | 0.217 | 0.169 | 0.304 | 0.265 | 0.364 | 0.180 | 0.292 | 0.106 | 0.236 | 0.080 | 0.189 | 0.090 | 0.203 |
| | 37.5% | 0.064 | 0.174 | 0.072 | 0.180 | 0.084 | 0.196 | 0.120 | 0.230 | 0.220 | 0.347 | 0.296 | 0.382 | 0.215 | 0.318 | 0.124 | 0.258 | 0.102 | 0.212 | 0.109 | 0.222 |
| | 50% | 0.097 | 0.204 | 0.107 | 0.216 | 0.102 | 0.215 | 0.141 | 0.248 | 0.293 | 0.402 | 0.334 | 0.404 | 0.257 | 0.347 | 0.165 | 0.299 | 0.133 | 0.240 | 0.137 | 0.248 |
| | Avg | 0.061 | 0.164 | 0.069 | 0.173 | 0.078 | 0.187 | 0.115 | 0.224 | 0.202 | 0.329 | 0.284 | 0.373 | 0.201 | 0.306 | 0.117 | 0.246 | 0.094 | 0.201 | 0.103 | 0.214 |
| ETTh2 | 12.5% | 0.036 | 0.119 | 0.039 | 0.125 | 0.040 | 0.130 | 0.057 | 0.152 | 0.187 | 0.319 | 0.101 | 0.231 | 0.100 | 0.216 | 0.095 | 0.212 | 0.042 | 0.133 | 0.044 | 0.138 |
| | 25% | 0.037 | 0.124 | 0.044 | 0.135 | 0.046 | 0.141 | 0.061 | 0.158 | 0.166 | 0.279 | 0.115 | 0.246 | 0.127 | 0.247 | 0.137 | 0.258 | 0.049 | 0.147 | 0.050 | 0.149 |
| | 37.5% | 0.039 | 0.136 | 0.051 | 0.147 | 0.052 | 0.151 | 0.067 | 0.166 | 0.400 | 0.465 | 0.126 | 0.257 | 0.158 | 0.276 | 0.187 | 0.304 | 0.056 | 0.158 | 0.060 | 0.163 |
| | 50% | 0.047 | 0.147 | 0.059 | 0.158 | 0.060 | 0.162 | 0.073 | 0.174 | 0.602 | 0.572 | 0.136 | 0.268 | 0.183 | 0.299 | 0.232 | 0.341 | 0.065 | 0.170 | 0.068 | 0.173 |
| | Avg | 0.040 | 0.131 | 0.048 | 0.141 | 0.049 | 0.146 | 0.065 | 0.163 | 0.367 | 0.436 | 0.119 | 0.250 | 0.142 | 0.259 | 0.163 | 0.279 | 0.053 | 0.152 | 0.055 | 0.156 |
| ECL | 12.5% | 0.042 | 0.150 | 0.080 | 0.194 | 0.085 | 0.202 | 0.055 | 0.160 | 0.196 | 0.321 | 0.102 | 0.229 | 0.092 | 0.214 | 0.107 | 0.237 | 0.093 | 0.210 | 0.089 | 0.210 |
| | 25% | 0.057 | 0.171 | 0.087 | 0.203 | 0.089 | 0.206 | 0.065 | 0.175 | 0.207 | 0.332 | 0.121 | 0.252 | 0.118 | 0.247 | 0.120 | 0.251 | 0.097 | 0.214 | 0.096 | 0.220 |
| | 37.5% | 0.067 | 0.179 | 0.094 | 0.211 | 0.094 | 0.213 | 0.076 | 0.189 | 0.219 | 0.344 | 0.144 | 0.273 | 0.144 | 0.276 | 0.136 | 0.266 | 0.102 | 0.220 | 0.104 | 0.229 |
| | 50% | 0.080 | 0.190 | 0.101 | 0.220 | 0.100 | 0.221 | 0.091 | 0.208 | 0.235 | 0.357 | 0.160 | 0.293 | 0.175 | 0.305 | 0.158 | 0.284 | 0.108 | 0.228 | 0.113 | 0.239 |
| | Avg | 0.061 | 0.173 | 0.090 | 0.207 | 0.092 | 0.210 | 0.072 | 0.183 | 0.214 | 0.339 | 0.131 | 0.262 | 0.132 | 0.260 | 0.130 | 0.259 | 0.100 | 0.218 | 0.101 | 0.225 |
| Weather | 12.5% | 0.016 | 0.042 | 0.026 | 0.049 | 0.025 | 0.045 | 0.029 | 0.049 | 0.057 | 0.141 | 0.047 | 0.101 | 0.039 | 0.084 | 0.041 | 0.107 | 0.027 | 0.051 | 0.026 | 0.047 |
| | 25% | 0.021 | 0.044 | 0.028 | 0.052 | 0.029 | 0.052 | 0.031 | 0.053 | 0.065 | 0.155 | 0.052 | 0.111 | 0.048 | 0.103 | 0.064 | 0.163 | 0.029 | 0.056 | 0.030 | 0.054 |
| | 37.5% | 0.025 | 0.042 | 0.033 | 0.060 | 0.031 | 0.057 | 0.035 | 0.058 | 0.081 | 0.180 | 0.058 | 0.121 | 0.057 | 0.117 | 0.107 | 0.229 | 0.033 | 0.062 | 0.032 | 0.060 |
| | 50% | 0.030 | 0.055 | 0.037 | 0.065 | 0.034 | 0.062 | 0.038 | 0.063 | 0.102 | 0.207 | 0.065 | 0.133 | 0.066 | 0.134 | 0.183 | 0.312 | 0.037 | 0.068 | 0.037 | 0.067 |
| | Avg | 0.023 | 0.046 | 0.031 | 0.056 | 0.030 | 0.054 | 0.060 | 0.144 | 0.076 | 0.171 | 0.055 | 0.117 | 0.052 | 0.110 | 0.099 | 0.203 | 0.032 | 0.059 | 0.031 | 0.057 |

Table 10: Full Results of Anomaly Detection

| Methods | SMD | | | MSL | | | SMAP | | | SWaT | | | PSM | | |
|---|---|---|---|---|---|---|---|---|---|---|---|---|---|---|---|
| Metrics | Precision | Recall | F-score | Precision | Recall | F-score | Precision | Recall | F-score | Precision | Recall | F-score | Precision | Recall | F-score |
| PTE4TS | 91.23 | 84.54 | 87.76 | 91.64 | 87.34 | 89.44 | 92.56 | 63.48 | 75.31 | 93.42 | 96.11 | 94.75 | 99.23 | 96.21 | 97.70 |
| FPT | 88.89 | 84.98 | 86.89 | 82.00 | 82.91 | 82.45 | 90.60 | 60.95 | 72.88 | 92.20 | 96.34 | 94.23 | 98.62 | 95.68 | 97.13 |
| TimesNet | 87.91 | 81.54 | 84.61 | 89.54 | 75.36 | 81.84 | 90.14 | 56.40 | 69.39 | 90.75 | 95.40 | 93.02 | 98.51 | 96.20 | 97.34 |
| PatchTST | 87.26 | 82.14 | 84.62 | 88.34 | 70.96 | 78.70 | 90.64 | 55.46 | 68.82 | 91.10 | 80.94 | 85.72 | 98.84 | 93.47 | 96.08 |
| ETSformer | 87.44 | 79.23 | 83.13 | 85.13 | 84.93 | 85.03 | 92.25 | 55.75 | 69.50 | 90.02 | 80.36 | 84.91 | 99.31 | 85.28 | 91.76 |
| FEDformer | 87.95 | 82.39 | 85.08 | 77.14 | 80.07 | 78.57 | 90.47 | 58.10 | 70.76 | 90.17 | 96.42 | 93.19 | 97.31 | 97.16 | 97.23 |
| LightTS | 87.10 | 78.42 | 82.53 | 82.40 | 75.78 | 78.95 | 92.58 | 55.27 | 69.21 | 91.98 | 94.72 | 93.33 | 98.37 | 95.97 | 97.15 |
| DLinear | 83.62 | 71.52 | 77.10 | 84.34 | 85.42 | 84.88 | 92.32 | 55.41 | 69.26 | 80.91 | 95.30 | 87.52 | 98.28 | 89.26 | 93.55 |
| Stationary | 88.33 | 81.21 | 84.62 | 68.55 | 89.14 | 77.50 | 89.37 | 59.02 | 71.09 | 68.03 | 96.75 | 79.88 | 97.82 | 96.76 | 97.29 |
| Autoformer | 88.06 | 82.35 | 85.11 | 77.27 | 80.92 | 79.05 | 90.40 | 58.62 | 71.12 | 89.85 | 95.81 | 92.74 | 99.08 | 88.15 | 93.29 |
| Pyraformer | 85.61 | 80.61 | 83.04 | 83.81 | 85.93 | 84.86 | 92.54 | 57.71 | 71.09 | 87.92 | 96.00 | 91.78 | 71.67 | 96.02 | 82.08 |
| Anomaly Transformer | 88.91 | 82.23 | 85.49 | 79.61 | 87.37 | 83.31 | 91.85 | 58.11 | 71.18 | 72.51 | 97.32 | 83.10 | 68.35 | 94.72 | 79.40 |
| Informer | 86.60 | 77.23 | 81.65 | 81.77 | 86.48 | 84.06 | 90.11 | 57.13 | 69.92 | 70.29 | 96.75 | 81.43 | 64.27 | 96.33 | 77.10 |
| Reformer | 82.58 | 69.24 | 75.32 | 85.51 | 83.31 | 84.40 | 90.91 | 57.44 | 70.40 | 72.50 | 96.53 | 82.80 | 59.93 | 95.38 | 73.61 |
| LogTransformer | 83.46 | 70.13 | 76.21 | 73.05 | 87.37 | 79.57 | 89.15 | 57.59 | 69.97 | 68.67 | 97.32 | 80.52 | 63.06 | 98.00 | 76.74 |
| Transformer | 83.58 | 76.13 | 79.56 | 71.57 | 87.37 | 78.68 | 89.37 | 57.12 | 69.70 | 68.84 | 96.53 | 80.37 | 62.75 | 96.56 | 76.07 |

