# OpenReview forum: "PTE4TS: One Pre-Training Encoder is All Time Series Need"
_ICLR.cc/2025/Conference — ICLR 2025 Conference Withdrawn Submission_

### Official Review · Reviewer_ZeZX · 2024-10-30

**Soundness:** 2
**Presentation:** 3
**Contribution:** 1
**Rating:** 3
**Confidence:** 5

**Summary:**

This research proposes PTE4TS, a general pre-trained encoder specifically designed for time series analysis, aiming to enhance the performance of various time series analysis tasks. It introduces a masking method that combines point-level and patch-level processing techniques, along with a hybrid attention encoder, to address the diminishing expressive capabilities arising from the low-rank structures of bidirectional attention mechanisms.

**Strengths:**

1. This results seems promising in mainstream tasks such as time series classification, anomaly detection, prediction, and data imputation.
2. The writing is easy to follow.

**Weaknesses:**

1.The paper lacks ablation studies. This paper could benefit from more granular ablation studies focusing on the contributions of the attention mechanism optimization and mask optimization separately. Additionally, analyzing the mask types (point-level vs. patch-level vs. point+patch) would offer insights into their contribution to learning local and global temporal dependencies.

2.The paper lacks in-depth analysis and robust experimental results to support the issues it aims to address. The proposed combination of patch and point masking lacks robust theoretical explanation. Is the task of point masking truly meaningful? How do you justify its rationale (not based on intuition or final experimental results)? Full-rank attention seems like an understandable optimization, but how can it be demonstrated that low-rank attention is unacceptable in time series pretraining?

3.The final downstream tasks are limited to the domains included in the pre-training data and do not verify the transferability of the pre-trained model. Tasks from completely different domains (e.g., exchange rates for finance or healthcare data) could highlight its potential transferability and broaden the applicability of the proposed model.

4.Moreover, the paper does not consider the impact of model parameter scale on the comparative performance of different models. Considering the model's parameter scale, especially when compared to similar models like TimesNet, PatchTST, and FPT, would offer more robust comparative insights. It is essential to ensure a fair comparison to draw valid conclusions from the experiments.

5.What is the relation between 'the mixed self-supervised task' and 'the low-rank problem'? These two issues seem quite disconnected, and I don't see any intrinsic link between them.

6.There are several minor formatting issues, such as inconsistent capitalization (e.g., "Attention" line 227) and abbreviation usage, and typographical errors in tables (e.g., "F-score" table 7).

**Questions:**

1.I have significant concerns about the motivation behind the 'mixed self-supervised task.' The point-level information method, with its relatively low information density, appears to lack practical significance, which is why patch-level masking is generally preferred in time series analysis. Could you clarify why this point-level approach is beneficial in the context of time series? Please provide an explanation from both mathematical principles and detailed experimental results to support its utility and effectiveness.

2.Does low-rank attention have a substantial impact on time series analysis (TSA), particularly in relation to its effectiveness compared to more common high-rank approaches? What specific benefits does low-rank attention offer over the decoder-only architecture, especially in terms of representation capacity and computational efficiency for TSA tasks?

3.Please refer to Weakness 3. The current discussion on task limitations and downstream performance could benefit from more detailed justification. Could you elaborate on how this limitation affects the model's generalizability to unseen data and its robustness across various time series domains?

4.Does this method align with known scaling laws in model performance and resource efficiency? Specifically, how does the approach scale as model size increases, and is its performance consistent with theoretical expectations for larger models in time series analysis?

---

### Official Review · Reviewer_CQY2 · 2024-10-31

**Soundness:** 2
**Presentation:** 3
**Contribution:** 2
**Rating:** 3
**Confidence:** 5

**Summary:**

This paper presents a general pre-training model for various downstream time series tasks. Specifically, it introduces a self-supervised pre-training approach that combines point-level and patch-level masking strategies. Additionally, a hybrid attention encoder is proposed to address the low-rank structures inherent in bidirectional attention mechanisms. Experimental results across time series forecasting, classification, and anomaly detection tasks demonstrate the effectiveness of the proposed method.

**Strengths:**

1. The writing and organization of this paper are good, and the title is very engaging.

2. This paper focuses on the task of time series pre-training, and the paradigm of pre-training and fine-tuning using large time series datasets rather than fine-tuning LLMs is highly challenging. It can provide a valuable reference for future research.

3. This paper clearly discusses existing time series pre-training techniques and pre-training methods based on the masking mechanism.

**Weaknesses:**

1. The technical contributions of this paper lack reasonable justification. Specifically, the authors use experimental conclusions from the ETTh2 and Electricity datasets as the motivation for proposing the Mix Mask and MDMH attention approach. These conclusions appear to involve cherry-picking and may not be generalizable to most time series datasets. ETTh2 and Electricity are relatively small datasets and only apply to time series forecasting tasks, making it difficult to directly apply these experimental conclusions to time series classification and anomaly detection tasks. To enhance the validation and impact of the technical contributions, the authors could assess the generalizability of their experimental findings on the UCR 128 archive (https://www.cs.ucr.edu/%7Eeamonn/time_series_data_2018/) for time series classification and the UCR 250 archive (https://wu.renjie.im/research/anomaly-benchmarks-are-flawed/) for anomaly detection. Such evaluations would strengthen the paper’s contributions to the field.

2. The masked layer normalization proposed in this paper utilizes the mean and variance of the output data for the normalization process, which closely resembles the approach described in the existing literature [R1]. The authors should provide a detailed comparison, highlighting both the similarities and differences between the masked layer normalization introduced here and that in [R1], within the main text.

3. The authors propose the Mix Mask mechanism by combining point-level and patch-level masking for pre-training. However, they do not address whether incorporating point-level masking, rather than using patch-level masking alone, might considerably increase the model's runtime. Additionally, the authors should discuss the trade-offs between runtime costs and the performance improvements in downstream tasks resulting from the inclusion of point-level masking.

4. The experiments in this paper lack comparisons with the latest baseline methods across the three main tasks of forecasting [R2, R3, R4, R5], classification [R1, R6], and anomaly detection [R7]. Additionally, the experimental setup and metric selection on forecasting [R8] and anomaly detection [R9] tasks require further improvement.

[R1] NuTime: Numerically Multi-Scaled Embedding for LargeScale Time-Series Pre-training. TMLR, 2024.

[R2] Unified Training of Universal Time Series Forecasting Transformers. ICML, 2024.

[R3] SparseTSF: Modeling Long-term Time Series Forecasting with 1k Parameters. ICML, 2024.

[R4] MOMENT: A Family of Open Time-series Foundation Models. ICML, 2024.

[R5] A decoder-only foundation model for time-series forecasting. ICML, 2024.

[R6] Bake off redux: a review and experimental evaluation of recent time series classification algorithms. DMKD, 2024.

[R7] DCdetector: Dual Attention Contrastive Representation Learning for Time Series Anomaly Detection. KDD, 2023.

[R8] TFB: Towards Comprehensive and Fair Benchmarking of Time Series Forecasting Methods. VLDB, 2024.

[R9] A Survey on Time-Series Pre-Trained Models. TKDE, 2024.

**Questions:**

1. In Figure 1, why were only the ETTh2 and Electricity datasets chosen, while other time series forecasting datasets, as well as those used for time series classification and anomaly detection tasks, were not included? Additionally, what impact do the dataset sample sizes and the length of individual time series samples have on the conclusions presented in Figure 1?

2. For the forecasting task, the literature [R8] indicates that existing methods employ the technique of "deleting the last batch of data" during the training phase, leading to an unfair comparison of final results. How does the proposed method perform in terms of forecasting performance when compared to the experimental setup presented in [R8]?

3. For the classification task, the literature [R6] indicates that many existing deep learning methods only select a subset of the UEA/UCR datasets for classification validation, raising concerns about cherry-picking in their conclusions. Based on the findings of [R6], the existing methods HC-2, MR-H, and H-InceptionTime perform well across 112 UCR datasets. Can the proposed method achieve state-of-the-art performance on the 112 UCR datasets?

4. For the anomaly detection task, the literature [R9] selects various evaluation metrics, totaling 12 metrics, for a comprehensive comparison. Relying solely on F1-score, Precision, and Recall metrics may lead to unfair comparison results [R10, R11]. Based on the multiple evaluation metrics employed in [R9], can the proposed method achieve state-of-the-art performance?

5. How does the proposed pre-training method's runtime and computational resource requirement compare to the performance of [R2, R4, T5]?

6. The proposed MDMH attention appears to apply to video or recommendation sequences. What are its specific advantages in addressing time series modeling? How does the proposed attention mechanism compare to unidirectional attention mechanisms in terms of runtime efficiency?

[R10] Towards a rigorous evaluation of time-series anomaly detection. AAAI, 2022.

[R11] Volume under the surface: a new accuracy evaluation measure for time-series anomaly detection. VLDB, 2022.

---

### Official Review · Reviewer_pSgP · 2024-11-03

**Soundness:** 3
**Presentation:** 2
**Contribution:** 2
**Rating:** 5
**Confidence:** 3

**Summary:**

The authors present a new timeseries model, and the method they used to train it. Unlike many previously published models, this model was first pre-trained using a large corpus (UTSD-12G), before being finetuned for the various test datasets.

The model is based on a patching + transformer architecture. Unlike the usual encoder (bidirectional) and decoder (unidirectional) architectures, the transformer architecture is one where some heads are unidirectional in the positive time direction, while the others are unidirectional in the negative time direction. The authors claim that this architecture allows to both have full rank attention (unlike the encoder architecture) and to be able to handle imputation problems (unlike the decoder architecture).

The forecast quality is tested against a wide swath of recent models, and comes ahead by a slight margin.

**Strengths:**

1. The authors include an ablation experiment, to explain their choice of bidirectional attention instead of alternative possibilities.
2. While it has been done before by recent papers (ex: Chronos by Ansari et al. 2024), the use of pretraining timeseries forecasting models using large corpus is a new field which merits exploration.
3. The numerical results shows that the new method is state-of-the-art.
4. The model is not only tested on forecasting, but also on imputation and anomaly detection tasks.

**Weaknesses:**

1. There are two main contributions to this paper: pretraining and the new architecture. It is however quite hard to distinguish the contribution to the improved numerical results by both either of them. While there are results with the new models both with and without finetuning (see Table 6), those without finetuning are relegated to the appendix, making them hard to compare.
2. According to Table 6, the MDMH attention scheme is barely better than more traditional attention schemes. This raises the possibility that the improvements are mainly due to more careful hyperparameter selection.
3. There is no comparison of the complexity of the various models under consideration. In particular, their training or inference speeds, and their memory footprints.
4. Hyperparameters for the proposed model are missing.
5. The training procedure is not detailed, beside a reference to Wu et al (2022) and the fact that the test set of UTSD-12G was excluded from the training set.
6. The importance of having a full rank QK matrix seems overstated. The softmax will (almost surely) make the attention weight a full rank matrix anyway. What matters is what possible attention patterns the model can generate. On that front, the main advantage of the decoder approach is that the model don't have to explicitly handle the time ordering of tokens through the matrices Q and K. How much of a benefit this is is unknown to me. If prior work have explored it, then it should be cited.

**Questions:**

1. How much overlap is there between the training set and the various test sets? Is there any of the test set which is fully absent from the training set, and thus could be used as a test set for zero-shot forecasting?
2. What is the time frames from which the data of the various test sets come from? Is it possible that there is highly correlated data in the training set from the same time frames?

---

### Official Review · Reviewer_ZzA6 · 2024-11-04

**Soundness:** 2
**Presentation:** 2
**Contribution:** 2
**Rating:** 3
**Confidence:** 4

**Summary:**

PTE4TS is a pre-trained encoder designed for general TIME Series Analysis tasks including forecasting, imputation, anomaly detection, and classification. The authors propose three main innovations on top of the standard transformer encoder architecture:
1.  PTE4TS employs a unique patch-level masking strategy tailored for time series data. Instead of masking entire patches with zeros or random values, the authors use the mean values of visible patches and apply random shuffling within each masked patch. This approach introduces distortions to make the masked learning task more challenging, helping the model capture complex patterns in the data.
2. Authors observed certain weaknesses in the existing attention mechanisms: (i) unidirectional attention falls short on mask task, while bidirectional attention suffers from low-rank structure. Authors design a Mixed-Direction Multi-Head (MDMH) Attention mechanism. By splitting attention heads into forward and backward directions, this encoder retains a full-rank structure, enhancing the model’s ability to capture temporal dependencies across time series while still keeping the masking task intact.
3. Finally the authors introduce a Masked LayerNorm technique to handle padding effectively during training. By masking padding values when calculating the mean and variance, this approach reduces the noise from padded sequences, allowing the model to learn more accurate statistical properties of the time series data, thereby improving robustness across varying sequence lengths.

Experiments and Analysis:
The authors conduct comprehensive experiments across a wide range of TSA tasks, including classification, forecasting, imputation, and anomaly detection. Evaluations on these tasks demonstrate that PTE4TS achieves state-of-the-art or competitive results compared to existing deep learning models, outperforming existing methods on several benchmarks. Pre-training on the large Unified Time Series Dataset enables the model to generalize effectively across tasks, and it performs robustly even with limited fine-tuning data.

**Strengths:**

-  The hybrid attention mechanism, particularly the Mixed-Direction Multi-Head (MDMH) Attention, is well-motivated and effectively implemented. This approach directly addresses the low-rank limitation in bidirectional attention, offering a novel solution to enhance expressiveness in time series models.
- The paper is structured logically, with a clear flow from motivation to background, methodology, and experimental results. Each section builds on the previous one, making it easy for readers to understand the need for and design of PTE4TS. Complex concepts are supported by clear, informative figures that enhance the reader's understanding.
- The authors conduct a thorough set of experiments across a diverse range of TSA tasks, including classification, forecasting, anomaly detection, and imputation. The inclusion of multiple tasks demonstrates the versatility of PTE4TS.

**Weaknesses:**

- The experimental comparisons are limited largely to non-pretrained, deep-learning-based models, which may not provide a fully fair evaluation. I think in its current setting the only model pretrained among the baselines is Timer (authors can correct me if I am wrong). Since PTE4TS benefits from pretraining on a large dataset, it would be more meaningful to include more recent foundation models or pre-trained architectures in the comparisons. This addition would better demonstrate PTE4TS’s advantages relative to other large-scale models.

- The paper lacks an ablation study to isolate the contributions of each component in PTE4TS (such as the hybrid attention encoder, Masked LayerNorm, and enhanced patch-level masking). Without this, it’s challenging to determine whether the performance improvements stem from the proposed architectural innovations or are primarily due to pretraining on a large dataset. An ablation study could help clarify the relative importance of each component and provide a more rigorous evaluation of the proposed methods.

- The experiments primarily focus on quantitative performance comparisons, with little analysis to explain why PTE4TS outperforms the baselines. There’s no examination of what specific aspects of the model contribute to its effectiveness or if there are particular cases where it struggles. An analysis of model interpretability or attention patterns, for instance, could provide insights into its strengths and potential weaknesses, enhancing the understanding of the model’s behavior.

- While the paper dedicates significant attention to Masked LayerNorm, this component is a relatively straightforward weighted LayerNorm that excludes masked parts from computation. Of course simple is better when it works, but without an ablation study to demonstrate its specific contribution to model performance it raises questions about its actual significance.

Less important (no effect on the score):

- There are a lot of typos in the paper. I wont include all here but I think a single iteration with a spell checker would improve the quality of the paper. Some examples:
    - Page 1: "analysisLiu et al."
    - Page 6: "R^d_m(odel)" in the in text equation for the attention.
    - Page 7: The last 3 sentences of paragraph 1 in 4.1.1 is repeated again in the next paragraph.
    ...
- The attention mechanism needs more math notation. I think the use of FA and BA to describe the forward and backward attention is too abstract. It would be more helpful if the math notation could be more specific to the actual equations in the paper.

- The masking part is not directly intuitive for me. After setting the masked portion of the data to the mean value, why do we need to shuffle the values within the patch? Are not they all the same, i.e. the mean value of the visible values?

**Questions:**

- Since PTE4TS benefits from pretraining on a large dataset, have the authors considered including more foundation models or other pretrained architectures as baselines?

- Could the authors conduct an ablation study to isolate the contributions of each component in PTE4TS (e.g.,pretraining on a large dataset, hybrid attention encoder, Masked LayerNorm, and enhanced patch-level masking)? Such an analysis would clarify whether the performance gains come from the proposed architecture or primarily from the large pretraining dataset.

- Can the authors provide an analysis to explain why PTE4TS performs better than the baselines? Insights into which aspects of the model contribute most to its performance or an examination of its interpretability would help deepen understanding of its behavior.

- In the masking technique, the masked portions of the data are set to the mean value and then shuffled within each patch. Could the authors clarify the purpose of shuffling these values if they are already set to the same mean value?

---

### Note · Authors · 2024-11-12

I have read and agree with the venue's withdrawal policy on behalf of myself and my co-authors.